# PROMPT TUNING WITH PROMPT-ALIGNED GRADIENT FOR VISION-LANGUAGE MODELS

## ABSTRACT

Thanks to the large pre-trained vision-language models (VLMs) like CLIP (Radford et al., 2021), we can craft a zero-shot classifier by discrete prompt design, *e.g.*, the confidence score of an image being "`[CLASS]`" can be obtained by using the VLM provided similarity between the image and the prompt sentence "`a photo of a [CLASS]`". Furthermore, prompting shows great potential for fast adaptation of VLMs to downstream tasks if we fine-tune the soft prompts with few samples. However, we find a common failure that improper fine-tuning or learning with extremely few-shot samples may even under-perform the zero-shot prediction. Existing methods still address this problem by using traditional anti-overfitting techniques such as early stopping and data augmentation, which lack a principled solution specific to prompting. In this paper, we present Prompt-aligned Gradient, dubbed `ProGrad` to prevent prompt tuning from forgetting the general knowledge learned from VLMs. In particular, `ProGrad` only updates the prompt whose gradient is aligned (or non-conflicting) to the general knowledge, which is represented as the optimization direction offered by the pre-defined prompt predictions. Extensive experiments demonstrate the stronger few-shot generalization ability of `ProGrad` over state-of-the-art prompt tuning methods. Codes and theoretical proof are in Appendix.

## 1 INTRODUCTION

After seeing and reading countless image-text association pairs, large and deep vision-language models (VLM) (Radford et al., 2021; Jia et al., 2021) can memorize the **general knowledge** (a.k.a. encyclopedic knowledge) about what visual patterns correspond to what textual sequence and vice versa. Thanks to the powerful language modeling of VLMs, we can establish a communication channel in human-readable natural language, *i.e.*, **prompt** (Liu et al., 2021a; Yao et al., 2021; Jin et al., 2022), to query the general knowledge. Prompting bridges the interface gap between the pre-trained and downstream tasks (*e.g.*, regression *vs.* classification) without the need for additional fine-tuning adaptation. For example, we can craft a concrete prompt—"`a photo of a [CLASS]`"—to achieve zero-shot image classification: by using the popular vision-language model CLIP (Radford et al., 2021), we input the image to the vision end and the prompt sentence to the language end, then obtain a vision-language similarity as the confidence score of classifying the image as "`[CLASS]`".

In practice, the prompt-based zero-shot image classification is not accurate because the hand-crafted prompt may not be the most machine-favorable (*e.g.*, "`this is a picture of`" could be more grammatically prevailing in VLM training), or not specific to the downstream domain (*e.g.*, "`a photo of a person doing`" is better in action recognition) (Radford et al., 2021). Recently, prompt tuning or prefix tuning (Lester et al., 2021; Liu et al., 2021b; Zhou et al., 2021; 2022) has been proposed to replace the hand-crafted prompt with a set of tunable word embedding vectors, which do not have to be translatable back to human-readable words.

Yet, prompt tuning is still as tricky as conventional fine-tuning: as the training continues, the generalization ability may decrease and even under-perform the zero-shot baseline. As shown in Figure 1(a&b), the prompt tuning method CoOp (Zhou et al., 2021) achieves the best results via early stopping, and its accuracies heavily drop by at most 4% when the training continues. Besides, Figure 1(c&d) show that CoOp underperforms zero-shot CLIP without augmentation or enough samples from downstream tasks. To the best of our knowledge, existing methods still rely on the

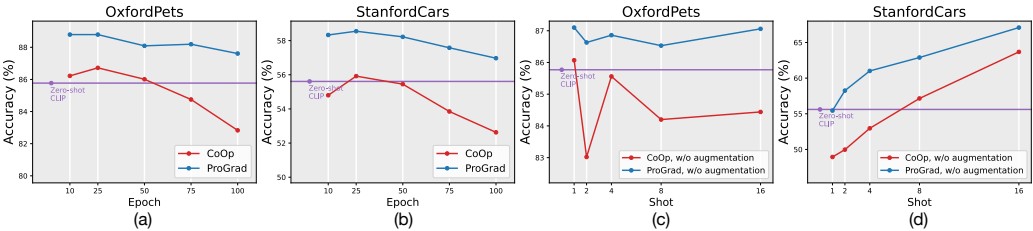

Figure 1: Comparison of Zero-shot CLIP, CoOp, and our `ProGrad` on Stanford Cars and OxfordPets datasets. (a)&(b): Given 1 shot training sample, CoOp's performance severely drops and underperforms zero-shot CLIP by large margins when the training continues. (c)&(d): CoOp may fail to improve CLIP without data augmentation or plenty of samples.

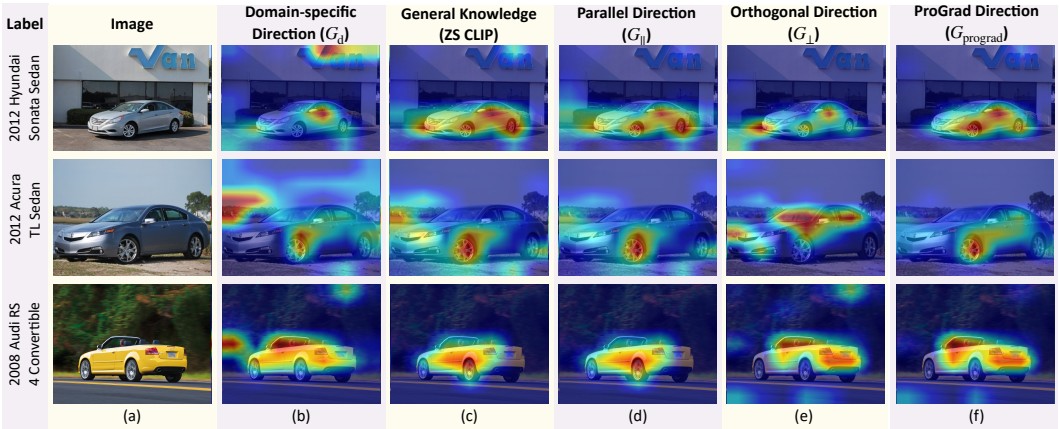

Figure 2: Comparisons of Grad-CAM visualization for prompt tuning methods using different gradient strategies on Stanford Cars Datasets. (b) Model trained with $G_d$. (c) Zero-shot CLIP model. (d) Model trained with $G_\parallel$. (e) Model trained with $G_\perp$. (f) Model trained with $G_{prograd}$.

conventional anti-overfitting techniques such as early stopping and data augmentation (Zhou et al., 2021; 2022; Gao et al., 2021b; Qin & Joty, 2022a), which lacks a principled solution to the nature of improper prompt tuning. Furthermore, the Grad-CAM visualization results indicate that the fine-tuned prompt starts to mislead the VLM to forget the general knowledge that the classification should at least focus on the foreground object but not the background. Comparing CoOp (Figure 2(b)) with zero-shot CLIP (Figure 2(c)), we find that the CoOp model distracts its attention to the background, while CLIP mainly focuses on the foreground object. These results demonstrate the over-fitting risk of existing prompt tuning strategies, especially when the number of training samples is extremely limited (*e.g.*, 1 or 2).

To this end, we present a novel prompt tuning method called Prompt-aligned Gradient (`ProGrad`) to overcome the improperly biased tuning for CLIP. The principle of `ProGrad` is to regularize each tuning step not to conflict with the general knowledge offered by the original prompt, *e.g.*, the zero-shot CLIP predictions. Specifically, we measure the general knowledge direction $G_g$ using the gradient of Kullback–Leibler (KL) divergence between the predictions of the zero-shot prompted CLIP and the few-shot fine-tuned model, which we name as **general direction**. Similarly, we compute the domain-specific knowledge direction $G_d$ using the gradient of cross-entropy between the ground-truth and the few-shot fine-tuned model, dubbed domain-specific direction. We decompose the domain-specific direction $G_d$ into: 1) a vector $G_\perp$ orthogonal to the general direction, which denotes the non-conflicting domain-specific knowledge; and 2) a vector $G_\parallel$ parallel to the general direction, which denotes the general knowledge. Note that the first gradient component does NOT override the general direction as any two orthogonal vectors can be transformed into two non-conflicting base vectors. For the second component, it must be one of the two directions: 1) the same of the general direction, which indicates that the update is aligned to the general knowledge, and 2) the opposite of general direction, indicating a conflicting update that should be discarded to avoid forgetting. Overall, in each iteration, `ProGrad` only updates the parameters in the prompt-aligned direction that has an

acute angle to the general direction. Compared to CoOp and CLIP, both $G_g$ and $G_\perp$ (Figure 2(d&e)) help to regularize the model to focus on the foreground, and our `ProGrad` (Figure 2(f)) can further improve the visual response.

Following CLIP, CoOp and CoCoOp (Zhou et al., 2022), we evaluate our ProGrad under the few-shot learning, domain generalization, base-to-new generalization and cross-dataset transfer settings over 11 image classification benchmark datasets, covering generic object classification, fine-grained image recognition, action classification. In summary, our ProGrad achieves: 1) clear improvement compared to CoOp over all of the 11 datasets; 2) clear improvement on the harmonic mean of base-class and new-class accuracies on all 11 datasets compared to CoOp and CoCoOp, and 3) clear improvement on both the source and target datasets of the domain generalization.

## 2 RELATED WORK

**Fine-tuning for VLMs.** It is for the VLM adaptation to various downstream tasks, *e.g.*, visual question answering (Kim et al., 2021; Tan & Bansal, 2019), visual grounding (Yao et al., 2021), image retrieval (Lu et al., 2019), semantic segmentation (Rao et al., 2021) and image classification (Zhou et al., 2021; 2022). We focus on image classification task. Conventional "pre-train then fine-tune" paradigm that plugs in an additional classifier on top of visual backbone and trained on downstream data has been widely-adopted, *e.g.*, Linear Probe (Radford et al., 2021). CLIP-Adapter (Gao et al., 2021a) and Tip-Adapter (Zhang et al., 2021) add vision and language feature adapter to boost conventional fine-tuning results. Recently, NLP community presents a novel fine-tuning paradigm named "prompt-based learning", which is formulated as a "fill-in-the-blank" cloze test, and fine-tunes the prompt to maximize the ground-truth token (Lester et al., 2021; Liu et al., 2021b). In CV community, CoOp (Zhou et al., 2021) uses a continuous prompt optimization from downstream data instead of hand-craft design. CoCoOp (Zhou et al., 2022) further extends CoOp by learning image conditional prompt rather than a static one to improve generalization to unseen classes. ProDA (Lu et al., 2022) adapts VLMs to downstream classification tasks by learning a prompt distribution over the output embedding space. VPT (Derakhshani et al., 2022) introduces variational prompt tuning by combining a base learned prompt with a residual vector sampled from a instance-specific underlying distribution. Our proposed `ProGrad` follows the line of *prompt-based learning* to improve both few-shot classification performance and generalization ability by aligning the gradient to general direction, without model structure modification or tuning the pre-trained model parameters.

**Knowledge Transfer.** Forgetting mitigation by knowledge distillation or memory replay is widely deployed in incremental learning (Liu et al., 2020; Rebuffi et al., 2017; Qin & Joty, 2022b; Riemer et al., 2018; Hu et al., 2021). However, prompt-based fine-tuning is fundamentally different from incremental learning: the former assumes that VLMs have already captured all the knowledge needed in downstream tasks and the goal is to compose a domain-specific query, whereas the latter assumes that the knowledge is yet to be sufficient. In addition, incremental learning requires old data from memory storage while our prompt-based learning method has no access to the pre-trained data. For example, OGD (Farajtabar et al., 2020) projects the gradients from new classes to the orthogonal direction of the gradients of previous tasks. However, as we have no access to pre-training process, the requirement of OGD to store the gradients of old tasks is not possible for prompt tuning. Moreover, OGD alters the gradients of downstream tasks even in non-conflicting scenarios which potentially results in sub-optimal performance for downstream tasks. Another related field that leverages gradient matching to transfer knowledge is domain generalization (Shi et al., 2022; Rame et al., 2021) and multi-task learning (Sener & Koltun, 2018; Yu et al., 2020). However, their methods are not directly applicable in prompt tuning whose transfer direction is only from general to downstream. In Appendix, we will show how their methods fail in several ablative studies.

## 3 METHODOLOGY

In this section, we introduce the preliminary concepts of hand-crafted prompt-based zero-shot inference, prompt-based learning, and present our proposed Prompt-aligned Gradient solution to align the domain knowledge with general knowledge for few-shot generalization.

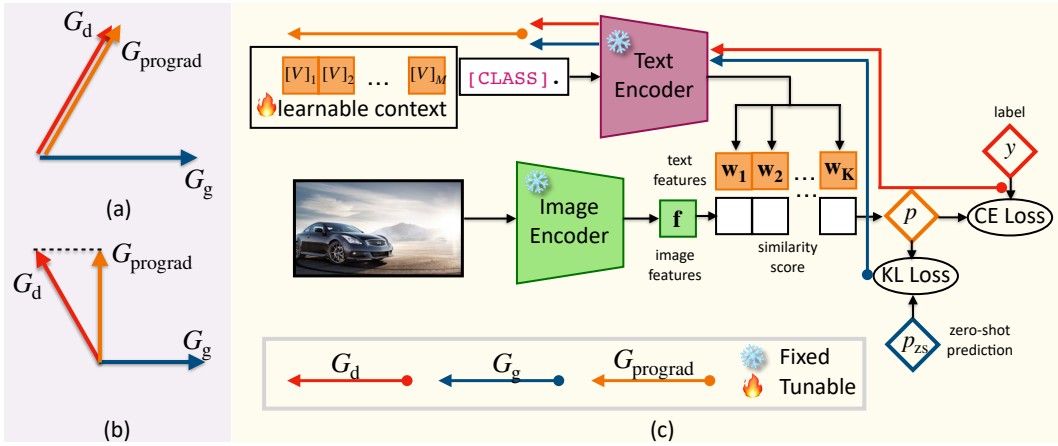

Figure 3: (a) If $G_\mathrm{d}$ is aligned with $G_\mathrm{g}$, we set $G_\mathrm{prograd}$ as $G_\mathrm{d}$. (b) If $G_\mathrm{d}$ conflicts with $G_\mathrm{g}$ (*i.e.*, their angle is larger than $90°$), we set $G_\mathrm{prograd}$ as the projection of $G_\mathrm{d}$ on the vertical direction of $G_\mathrm{g}$. (c) Training pipeline of our `ProGrad`. Only the context vectors are learnable.

## 3.1 PRELIMINARIES

**Contrastive language-image pre-training (CLIP)** (Radford et al., 2021) adopts a contrastive language-image pre-training paradigm on tremendous pairs of images with natural language descriptions. For contrastive learning, the associated image and sentences are taken as the positive samples, while the non-associated pairs are regarded as negative samples. The contrastive objective maximizes the similarity of positive pairs while minimize the similarity of negative pairs.

**Zero-shot transfer inference** adapts the pre-trained CLIP model to downstream tasks without fine-tuning the model. Taking image classification as an example, zero-shot transfer is enabled by formulating the classification task as an image-text matching problem, where the text is obtained by extending the "`[CLASS]`" name using a template like "`a photo of a [CLASS].`". CLIP (Radford et al., 2021) finds that such a simple template narrows the distribution gap to pre-training text inputs. The image-class matching score is measured based on the cosine similarity $< \boldsymbol{w}_i, \boldsymbol{f} >$ between the image feature $\boldsymbol{f}$ and the class-extended text feature $\boldsymbol{w}_i$ for $i$-th class. The image feature $\boldsymbol{f}$ for image $\boldsymbol{x}$ is extracted by the image encoder, while the text feature $\boldsymbol{w}_i$ for $i$-th class is obtained by feeding the prompt description into the text encoder. The probability for $i$-th class is obtained as

$$p_\mathrm{zs}(\boldsymbol{w}_i|\boldsymbol{x}) = \frac{\exp(< \boldsymbol{w}_i, \boldsymbol{f} > /\tau)}{\sum_{j=1}^{K} \exp(< \boldsymbol{w}_j, \boldsymbol{f} > /\tau)}, \tag{1}$$

where $K$ denotes the number of classes, and $\tau$ is a temperature learned by CLIP.

**Prompt-based learning** further strengths the transferring ability of the CLIP model and avoids prompt engineering by automatically learning the prompt given few samples from the downstream task. Different from the zero-shot transfer that used a fixed hand-craft prompt, CoOp (Zhou et al., 2021) constructs and fine-tunes a set of $M$ continuous context vectors $\boldsymbol{v} = \{\boldsymbol{v}_1, \boldsymbol{v}_2, ..., \boldsymbol{v}_M\}$ as the turnable prompt. Specifically, the prompt $\boldsymbol{t}_i = \{\boldsymbol{v}_1, \boldsymbol{v}_2, ..., \boldsymbol{v}_M, \boldsymbol{c}_i\}$ combines the learnable context vectors $\boldsymbol{v}$ and the class token embedding $\boldsymbol{c}_i$, and is fed to the text encoder $g(\cdot)$. CoOp optimizes the static context vectors $\boldsymbol{v}$ by minimizing the negative log-likelihood of the ground-truth token:

$$\mathcal{L}_\mathrm{ce}(\boldsymbol{v}) = -\sum_i \boldsymbol{y}_i \log p(\boldsymbol{t}_i|\boldsymbol{x}), \quad p(\boldsymbol{t}_i|\boldsymbol{x}) = \frac{\exp(< g(\boldsymbol{t}_i), \boldsymbol{f} > /\tau)}{\sum_{j=1}^{K} \exp(< g(\boldsymbol{t}_j), \boldsymbol{f} > /\tau)}, \tag{2}$$

where $\boldsymbol{y}$ denotes the one-hot ground-truth annotation and $K$ denotes the number of classes.

## 3.2 PROMPT-ALIGNED GRADIENT

As we introduced in Section 1, CoOp faced a challenge that the transfer performance drops when the number of annotations is very limited (*e.g.*, one per class), even underperforms the zero-shot transfer.

Also, CoOp heavily relies on anti-overfitting techniques such as early stopping and data augmentation. To overcome the over-fitting challenge, we propose an effective and efficient fine-tuning paradigm `ProGrad` to align the few-shot downstream knowledge with the large-scale general knowledge.

Motivated by the success of knowledge distillation (Phuong & Lampert, 2019; Hinton et al., 2015) in knowledge transfer, we leverage the zero-shot CLIP predictions as the general knowledge, and compare the fine-tuned predictions with the general knowledge to regularize the gradient direction. Specifically, we obtain the domain-specific direction by calculating the cross-entropy $\mathcal{L}_{\mathrm{ce}}(\boldsymbol{v})$ between the model prediction $p(\boldsymbol{t}_i|\boldsymbol{x})$ and the ground-truth $\boldsymbol{y}$ according to Eq. (2), and the general knowledge direction based on the Kullback-Leibler (KL) divergence between $p(\boldsymbol{t}_i|\boldsymbol{x})$ and the zero-shot CLIP prediction $p_{\mathrm{zs}}(\boldsymbol{w}_i|\boldsymbol{x})$:

$$\mathcal{L}_{\mathrm{kl}}(\boldsymbol{v}) = -\sum_i p_{\mathrm{zs}}(\boldsymbol{w}_i|\boldsymbol{x}) \log \frac{p(\boldsymbol{t}_i|\boldsymbol{x})}{p_{\mathrm{zs}}(\boldsymbol{w}_i|\boldsymbol{x})}. \tag{3}$$

We denote the gradients of $\mathcal{L}_{\mathrm{kl}}(\boldsymbol{v})$ and $\mathcal{L}_{\mathrm{ce}}(\boldsymbol{v})$ as $\boldsymbol{G}_{\mathrm{g}} = \nabla_{\boldsymbol{v}} \mathcal{L}_{\mathrm{kl}}(\boldsymbol{v})$ and $\boldsymbol{G}_{\mathrm{d}} = \nabla_{\boldsymbol{v}} \mathcal{L}_{\mathrm{ce}}(\boldsymbol{v})$, respectively. The relations between $\boldsymbol{G}_{\mathrm{g}}$ and $\boldsymbol{G}_{\mathrm{d}}$ are two-fold. (1) Their angle is smaller than 90°(Figure 3(a)), which indicates that the optimization direction of few-shot downstream knowledge does not conflict with general knowledge. In this case, we safely set the updated gradient direction $\boldsymbol{G}_{\mathrm{prograd}}$ as $\boldsymbol{G}_{\mathrm{d}}$. (2) Their angle is larger than 90°(Figure 3(b)), which indicates that the few-shot downstream knowledge conflicts with general knowledge. In other words, optimizing the context vectors following $\boldsymbol{G}_{\mathrm{d}}$ will lead to the forgetting of the pre-trained general knowledge. In this case, we project the $\boldsymbol{G}_{\mathrm{d}}$ to the orthogonal direction of $\boldsymbol{G}_{\mathrm{g}}$ to optimize the model for classification, which avoids increasing the KL loss. Our `ProGrad` strategy is mathematically formulated as:

$$\boldsymbol{G}_{\mathrm{prograd}} = \begin{cases} \boldsymbol{G}_{\mathrm{d}}, & \text{if } \boldsymbol{G}_{\mathrm{d}} \cdot \boldsymbol{G}_{\mathrm{g}} \geq 0 \\ \boldsymbol{G}_{\mathrm{d}} - \lambda \cdot \frac{\boldsymbol{G}_{\mathrm{d}} \cdot \boldsymbol{G}_{\mathrm{g}}}{\|\boldsymbol{G}_{\mathrm{g}}\|^2} \boldsymbol{G}_{\mathrm{g}}, & \text{otherwise.} \end{cases} \tag{4}$$

Fig 3(c) illustrates the pipeline of our `ProGrad`. Instead of updating the context vectors using $\boldsymbol{G}_{\mathrm{d}}$ in CoOp (Zhou et al., 2021), we optimize the context vectors using $\boldsymbol{G}_{\mathrm{prograd}}$, which prevent the gradient direction from overfitting to few-shot downstream samples. We further introduce $\lambda$ in Eq. (4) to generalize the formulation, which can flexibly control the strength of general knowledge guidance in applications. In particular, $\lambda = 1$ denotes projecting $\boldsymbol{G}_{\mathrm{d}}$ to the orthogonal direction of $\boldsymbol{G}_{\mathrm{g}}$ (Figure 3(b)), while setting $\lambda = 0$ makes `ProGrad` degenerate to CoOp, *i.e.*, CoOp is a special case of our strategy. We include the detailed analysis of $\lambda$ in Appendix.

**Generalization Error Analysis**. We further theoretically analyze the generalization error of our `ProGrad`. Here, we provide a sketch proof and include the detailed justification in Appendix. Our `ProGrad` keeps the optimal value $\mathcal{L}_{\mathrm{kl}}$ of the pre-trained domain when optimizing the empirical risk on the downstream domain. The model $\hat{f}_{\mathrm{prograd}}$ learned by such update rule can be viewed as optimizing the empirical risk on pre-trained and downstream domains (Yu et al., 2020):

$$\hat{f}_{\mathrm{prograd}} = \underset{f \in \mathcal{F}}{\arg\min} \, \hat{\mathcal{R}}_{(d+p)}(f) = \underset{f \in \mathcal{F}}{\arg\min} \, \hat{\mathcal{R}}_d(f) + \hat{\mathcal{R}}_p(f), \tag{5}$$

where $\mathcal{F}$ is the function class, and $\mathcal{R}(\cdot)$ and $\hat{\mathcal{R}}(\cdot)$ denote the expected risk and empirical risk. We bound the generalization error of `ProGrad` by virtue of *Rademacher Complexity* (Bartlett & Mendelson, 2002) and the Theorem 6.2 in (Zhang et al., 2012). The detailed proof is in Appendix.

**Theorem 1** *Let $\mathbf{X}_1^{N_d} = \{\mathbf{x}_n^{(d)}\}_{n=1}^{N_d}$ and $\mathbf{X}_1^{N_p} = \{\mathbf{x}_n^{(p)}\}_{n=1}^{N_p}$ be two set of i.i.d. samples drawn from the downstream domain $\mathcal{D}_d$ and the pre-trained domain $\mathcal{D}_p$. Then for any $\epsilon > 0$, we have with probability at least $1 - \epsilon$,*

$$\mathcal{R}_d(\hat{f}_{prograd}) \leq \hat{\mathcal{R}}_{(d+p)}(\hat{f}_{prograd}) + \frac{1}{2} \gamma_{\mathcal{F}}(D, P) + \mathfrak{R}_p(\mathcal{F}) + \mathfrak{R}_d(\mathcal{F})$$
$$+ \frac{3}{2} \sqrt{\frac{\ln(4/\epsilon)}{2N_d}} + \frac{3}{2} \sqrt{\frac{\ln(4/\epsilon)}{2N_p}} + \frac{1}{2} \sqrt{\frac{\ln(4/\epsilon)}{2} \left( \frac{1}{N_d} + \frac{1}{N_p} \right)}, \tag{6}$$

*where $\gamma_{\mathcal{F}}(D, P)$ is the integral probability metric (Müller, 1997) that measures the difference between the distribution of pre-trained domain and the downstream domain, $\mathfrak{R}_d(\mathcal{F})$ and $\mathfrak{R}_p(\mathcal{F})$ are the Rademacher complexity of $\mathcal{F}$.*

Note that the bound of $\mathfrak{R}(\mathcal{F})$ is inversely proportional to the number of training samples. Theorem 1 shows that the generalization error $\mathcal{R}_d(\hat{f}_{\text{prograd}})$ is bounded by the empirical training risk $\hat{\mathcal{R}}_{(d+p)}(\hat{f}_{\text{prograd}})$, the two domain gap $\gamma_{\mathcal{F}}(D, P)$ and the estimation error. The empirical training risk can be minimized to arbitrary small value when using deep models with high capacity. The estimation error that related to $N_p$ asymptotically tends to 0 as the sample size $N_p$ tends to infinity. Thanks to the large amount of pretrained samples $N_p$, we can approximate the generalization error bound as

$$\mathcal{R}_d(\hat{f}_{\text{prograd}}) \leq \frac{1}{2}\gamma_{\mathcal{F}}(S, P) + \mathfrak{R}_d(\mathcal{F}) + \frac{3}{2}\sqrt{\frac{\ln(4/\epsilon)}{2N_d}} + \frac{1}{2}\sqrt{\frac{\ln(4/\epsilon)}{2}\frac{1}{N_d}}. \tag{7}$$

Similarly, we have the generalization error for CoOp $\hat{f}_{\text{coop}}$ as

$$\mathcal{R}_d(\hat{f}_{\text{coop}}) \leq 2\mathfrak{R}_d(\mathcal{F}) + 3\sqrt{\frac{\ln(4/\epsilon)}{2N_d}} + \sqrt{\frac{\ln(4/\epsilon)}{2}\frac{1}{N_d}}. \tag{8}$$

Under the assumption that the gap between pre-trained and downstream domains $\gamma(P, D)$ is small, the estimation error bound of $\mathcal{R}_d(\hat{f}_{\text{coop}})$ is at least two times greater than $\mathcal{R}_d(\hat{f}_{\text{prograd}})$. Considering that $N_d$ is typically very small in few-shot setting, our `ProGrad` model $\hat{f}_{\text{prograd}}$ achieves a much lower error bound than conventional fine-tuning model like CoOp $\hat{f}_{\text{coop}}$.

## 4 EXPERIMENTS

### 4.1 DATASETS AND IMPLEMENTATION DETAILS

We follow CLIP, CoOp and CoCoOp to validate the effectiveness of our `ProGrad` on four settings: (1) few-shot classification, (2) domain generalization, (3) base-to-new generalization, (4) cross-dataset transfer. We report and discuss the results of cross-dataset transfer in Appendix D.1

**Datasets.** For evaluations of few-shot learning and base-to-new generalization, we follow CLIP and CoOp to use 11 image classification datasets, *i.e.*, ImageNet (Deng et al., 2009) and Caltech101 (Fei-Fei et al., 2004) for generic object classification, OxfordPets (Parkhi et al., 2012), StanfordCars (Krause et al., 2013), Flowers102 (Nilsback & Zisserman, 2008), Food101 (Bossard et al., 2014) and FGVCAircraft (Maji et al., 2013) for fine-grained image recognition, EuroSAT (Helber et al., 2019) for satellite image classification, UCF101 (Soomro et al., 2012) for action classification, DTD (Cimpoi et al., 2014) for texture classification, and SUN397 (Xiao et al., 2010) for scene recognition. For domain generalization, we use ImageNet as the source dataset and select ImageNetV2 (Recht et al., 2019), ImageNet-Sketch (Wang et al., 2019), ImageNet-A (Hendrycks et al., 2021b) and ImageNet-R (Hendrycks et al., 2021a) as the target datasets.

**Training Details.** For few-shot learning, following CoOp and CLIP, all models are trained with {1, 2, 4, 8, 16} shots respectively then evaluated on the full test split. For domain generalization and base-to-new generalization, we evaluate 4-shot performance, which justifies the robustness under low-shots condition. All results of learning-based models are averaged over three random seeds. Unless otherwise stated, we adhere to CoOp to use ResNet-50 (He et al., 2016) as the backbone of image encoder. Following CoOp and CoCoOp, the length of context tokens $M$ is set to 16 for few-shot classification and $M = 4$ for base-to-new, domain generalization and cross-dataset transfer. We follow the same training epochs, training schedule and the data augmentation settings in CoOp. $\lambda$ is set to 1 by default, except that $\lambda$ is set to 0.8 for 16 shots. Please refer to Appendix for more details.

**Baselines.** We compare `ProGrad` with 4 baselines: (1) Zero-shot CLIP (2) Linear probe CLIP (3) CoOp and (4) CoCoOp. Although our method can beat some other fine-tune methods like CLIP-Adapter (Gao et al., 2021a), we mainly focus on comparing with *prompt-based learning* methods. The results of other fine-tuning methods are in Appendix.

### 4.2 FEW-SHOT CLASSIFICATION

**Setup.** We compare with two zero-shot CLIP models, *i.e.*, CLIP and CLIP++ stand for using single prompt and prompt ensembling respectively (Please refer to Appendix Section C for the templates

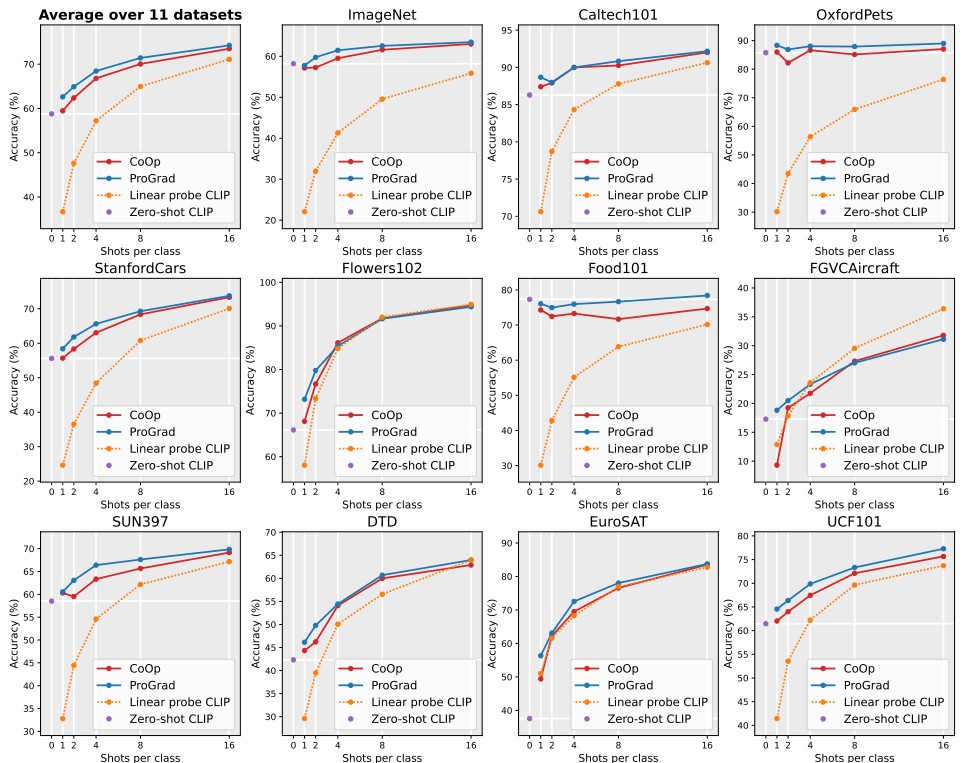

Figure 4: Accuracy (%) of few-shot learning on 11 datasets. The context length $M$ is set to 16.

of single and ensemble prompts). `ProGrad` and `ProGrad++` stand for using single prompt and prompt ensembling as general knowledge to implement ProGrad respectively. Note that we only use the hand-crafted prompt ensembling to generate $G_g$, which provides a more accurate general direction. Therefore, we still optimize a single prompt with 16 learnable tokens for `ProGrad++`.

Table 1 provides averaged accuracy over 11 datasets. Figure 4 illustrates the detailed comparisons over 11 datasets. Overall, `ProGrad` achieves clear advantages over baselines for all the few-shot settings on average performance. Specifically, `ProGrad` outperforms CoOp by 9.5%, 6.9% and 5.1% on FGVCAircraft, EuroSAT and Flowers102 given 1 shot, and the average improvement over 11 datasets is 3.2%. These results demonstrate the anti-overfitting ability of our

Table 1: Averaged accuracy (%) of few-shot learning on 11 datasets. $M = 16$.

| #shots | 0 | 1 | 2 | 4 | 8 | 16 |
|---|---|---|---|---|---|---|
| CLIP | 58.77 | - | - | - | - | - |
| CLIP++ | **59.38** | - | - | - | - | - |
| LP | - | 37.32 | 48.02 | 57.27 | 64.88 | 70.57 |
| CoOp | - | 59.46 | 62.37 | 66.79 | 70.04 | 73.49 |
| ProGrad | - | 62.61 | 64.90 | 68.45 | 71.41 | 74.28 |
| ProGrad++ | - | **63.06** | **65.28** | **68.71** | **71.80** | **75.03** |

`ProGrad` when the training samples are extremely limited. Furthermore, leveraging prompt ensembling can further explore the potential of `ProGrad`. From Table 1, with more accurate general knowledge offered by prompt ensembling, CLIP++ improves the zero-shot CLIP from $58.77\%$ to $59.38\%$; `ProGrad++` increases the accuracy of `ProGrad` from $74.28\%$ to $75.03\%$ at 16 shots.

### 4.3 DOMAIN GENERALIZATION

This setting evaluates the generalization ability of models on a target domain which is different from the source domain. Conventional fine-tuning on limited data from a specific domain may mislead the model to learn spurious correlations or in-distribution patterns, resulting in a biased model with under-performance in unseen domains. In contrast, zero-shot CLIP does not exploit such spurious correlations or patterns, since it is not fine-tuned on that distribution. Since our `ProGrad` uses the general knowledge from the pre-trained domain to regularize the fine-tuning on a specific distribution, our `ProGrad` is hopefully robust to the distribution shift. As shown in Table 2, despite the exposure

Table 2: Evaluation on robustness to distribution shift with different visual backbones.

(a) ResNet50

|  | Source | Target | | | |
| --- | --- | --- | --- | --- | --- |
|  | ImageNet | -V2 | -Sketch | -A | -R |
| CLIP | 58.18 | 51.34 | 33.32 | 21.65 | 56.00 |
| LP | 41.29 | 33.65 | 13.09 | 11.18 | 26.82 |
| CoOp | 61.34 | 53.81 | 32.83 | 22.08 | 54.62 |
| CoCoOp | 61.04 | 53.71 | 32.30 | 22.07 | 53.60 |
| ProGrad | **62.17** | **54.70** | **34.40** | **23.05** | **56.77** |

(b) ResNet101

|  | Source | Target | | | |
| --- | --- | --- | --- | --- | --- |
|  | ImageNet | -V2 | -Sketch | -A | -R |
| CLIP | 61.24 | 54.82 | 38.66 | 28.03 | 64.34 |
| LP | 47.01 | 38.46 | 19.09 | 16.33 | 39.43 |
| CoOp | 63.99 | 56.99 | 39.40 | 29.50 | 64.04 |
| CoCoOp | 63.59 | 56.98 | 39.16 | 29.09 | 64.14 |
| ProGrad | **64.98** | **57.86** | **40.53** | **30.13** | **65.61** |

(c) ViT-B/32

|  | Source | Target | | | |
| --- | --- | --- | --- | --- | --- |
|  | ImageNet | -V2 | -Sketch | -A | -R |
| CLIP | 62.00 | 54.75 | 40.82 | 29.59 | 66.01 |
| LP | 46.77 | 39.12 | 20.32 | 16.32 | 39.48 |
| CoOp | 64.74 | 56.59 | 40.03 | 31.10 | 64.54 |
| CoCoOp | 64.63 | 56.59 | 40.74 | 30.27 | 64.12 |
| ProGrad | **65.36** | **57.42** | **41.73** | **31.89** | **66.53** |

(d) ViT-B/16

|  | Source | Target | | | |
| --- | --- | --- | --- | --- | --- |
|  | ImageNet | -V2 | -Sketch | -A | -R |
| CLIP | 66.73 | 60.84 | 46.13 | 47.80 | 74.01 |
| LP | 54.70 | 45.57 | 28.20 | 22.47 | 44.12 |
| CoOp | 69.86 | 62.83 | 46.90 | 48.98 | 74.55 |
| CoCoOp | 70.13 | 63.05 | 46.48 | 49.36 | 73.80 |
| ProGrad | **70.45** | **63.35** | **48.17** | **49.45** | **75.21** |

(a) Failure cases analysis on EuroSAT  (b) Failure cases analysis on Oxford Flowers  (c) Failure cases analysis on UCF101

Figure 5: Distribution of samples that are mis-classified by ProGrad but correctly classified by CoOp.

to the source dataset, `ProGrad` clearly outperforms Zero-shot CLIP, CoOp and CoCoOp on all target datasets as well as the source dataset with ResNet-based and Transformer-based visual backbones.

## 4.4 BASE-TO-NEW GENERALIZATION

We follow (Zhou et al., 2022) to evaluate the generalization performance from seen classes to unseen classes. All the classes are equally divided into two groups, *i.e.*, base classes and new classes, and all the methods are only trained on base classes and tested on both base classes and novel classes. The harmonic mean of base-class and new-class accuracies is reported to evaluate the trade-off. Compared to CoOp and CoCoOp, `ProGrad` also generalizes well to the new classes. From Table 3, we observed

Table 3: Averaged accuracy (%) over 11 datasets for base-to-new generalization.

|  | Base | New | Harmonic |
| --- | --- | --- | --- |
| CLIP | 61.72 | 65.91 | 63.64 |
| CoOp | 71.96 | 61.26 | 65.58 |
| CoCoOp | 72.23 | 60.77 | 65.35 |
| ProGrad | **73.29** | **65.96** | **69.06** |

that `ProGrad` achieves the best average performance in terms of all metrics. In contrast, CoOp and CoCoOp fail in new classes: their performance is consistently worse than zero-shot CLIP. These results highlight that `ProGrad` has better generalizability to both seen and unseen classes. Please refer to Appendix for the results of 11 datasets.

## 4.5 FURTHER ANALYSIS

**Failure cases**. We further analyze the failure cases where `ProGrad` models predict incorrectly but CoOp gives right predictions. Specifically, we count the percentage of the failure cases that zero-shot CLIP models also fails in Figure 5. We found that a high proportion of the failure cases are also mis-classified by Zero-shot CLIP model (red bar in Figure 5). This observation indicates that the general direction $G_g$ generated by imprecise zero-shot general knowledge is detrimental to model generalization. As the number of samples increases, the downstream knowledge represented by $G_d$ becomes more accurate and unbiased. As expected, we observe that the red bar becomes larger.

**Conflict of knowledge.** `ProGrad` requires the updated gradient direction be acute to the general knowledge gradient directions. We explore how this constraint helps to defuse the conflicts of domain-

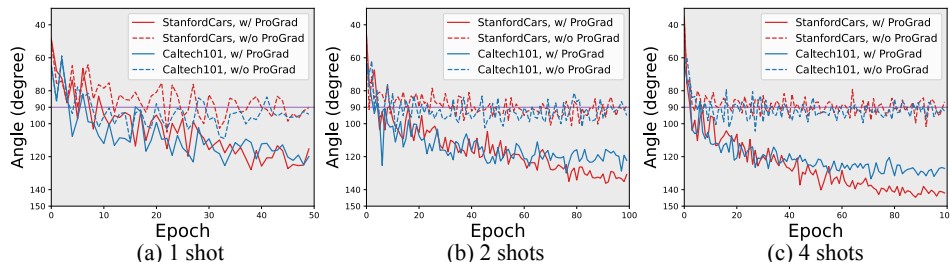

Figure 6: The angles between $G_d$ and $G_g$ during training on Caltech101 and StanfordCars.

specific and general knowledge by visualizing the angle between their representative gradients during training (angle between $G_d$ and $G_g$). As depicted in Figure 6, for the training without $G_{prograd}$, the angle between $G_d$ and $G_g$ converges to 90 degree due to the fact that "all high-dimensional random vectors are almost always orthogonal to each other" (Cai et al., 2013). Intuitively, without any constraint, the optimization direction $G_d$ is independent to the general direction, and the average angle would be around 90 degree (*i.e.*, orthogonal). In contrast, utilizing $G_{prograd}$ during training leads to the result that the angle finally converge to an obtuse angle. The reason is that $G_{prograd}$ intervenes the model to learn the downstream knowledge aligned with the general knowledge and leads to the insufficient learning of downstream knowledge that is incompatible with the general knowledge. As training stabilizes, $G_d$ struggles to learn the conflicting knowledge, reflecting an obtuse angle to the $G_g$. Thanks to `ProGrad`, we discard such conflicting knowledge to avoid forgetting.

**Comparison with conventional knowledge distillation.** Since our `ProGrad` utilizes the gradient direction of knowledge distillation loss as regularization, one may wonder whether our `ProGrad` is indeed conventional knowledge distillation. We answer this question by investigating whether a simple knowledge distillation (*i.e.*, $\mathcal{L}_{total} = \mathcal{L}_{ce} + \alpha \cdot \mathcal{L}_{kd}$) can achieve similar performance as our `ProGrad`. We repeated the few-shot experiments on 11 datasets with a variety of $\alpha$ and report the average results in Table 4. Overall, `ProGrad` outperforms KD for various few-shot settings. Although KD with small $\alpha \leq 1$ promotes CoOp in low-shot (*e.g.*, 1, 2 and 4 shots), the performance drops when number of shots is large (see 8 and 16 shots). These results indicate that our `ProGrad` works differently from KD and is more robust to the number of training samples.

Table 4: Comparison with knowledge distillation. Average accuracy (%) over 11 datasets.

| #shots | 1 | 2 | 4 | 8 | 16 |
|---|---|---|---|---|---|
| CoOp | 59.46 | 62.37 | 66.79 | 70.04 | 73.49 |
| KD, $\alpha = 0.25$ | 61.09 | 63.01 | 67.74 | 70.90 | 73.39 |
| KD, $\alpha = 0.5$ | 61.13 | 63.36 | 67.14 | 70.34 | 72.68 |
| KD, $\alpha = 1$ | 61.52 | 64.07 | 66.52 | 70.01 | 72.01 |
| KD, $\alpha = 2$ | 60.98 | 62.66 | 64.92 | 67.78 | 68.98 |
| KD, $\alpha = 4$ | 59.58 | 61.76 | 62.92 | 65.01 | 65.42 |
| ProGrad | **62.61** | **64.90** | **68.45** | **71.41** | **74.28** |

Table 5: Applying ProGrad to cosine classifier. Average accuracy (%) over 11 datasets.

| #shots | 1 | 2 | 4 | 8 | 16 |
|---|---|---|---|---|---|
| Cosine | 30.50 | 43.74 | 53.33 | 61.26 | 65.00 |
| + ProGrad | **32.29** | **46.14** | **55.18** | **62.05** | **66.47** |

**Applying `ProGrad` to conventional fine-tune paradigm.** We are also interested in whether `ProGrad` can be applied to the conventional "pre-train then fine-tune" paradigm. Specifically, we plug in an additional cosine classifier on top of the visual backbone and compare the performance by conducting the few-shot experiments. Table 5 shows that conventional fine-tuning can benefit from our `ProGrad`. The implementation details and the result of each dataset are provided in Appendix.

We also analyze the upper bound performance of ProGrad and effect of the hyper-parameter $\lambda$. The results and discussion are presented in Appendix D.3 and Appendix D.2

## 5 CONCLUSION

In this paper, we pointed out the over-fitting issues of existing prompt tuning methods for few-shot generalization, which heavily relies on early stopping and data augmentation to promote zero-shot inference. We proposed a prompt tuning method `ProGrad` that regularize each tuning step not to conflict with the general knowledge of the hand-crafted prompt. Experiments on few-shot classification, base-to-new generalization and domain generalization over 11 datasets demonstrate the effectiveness and efficiency of our `ProGrad`. In the future, we will explore how to apply `ProGrad` on other tasks like object detection and segmentation.

## ETHIC STATEMENT

No human subjects were involved during the research and developments of this work. All of our experiments were conducted on the standard benchmarks in the lab-based, controlled environment. Thus, due to the abstract nature of this work, it has minimal concerns regarding issues such as discrimination/bias/fairness, privacy, etc.

## REPRODUCIBILITY STATEMENT

In this paper, we conduct the experiments three times and report the mean and standard deviation (confident interval at 95%) values to alleviate the randomness of the starting seed. In Appendix Section D.4, we provide the full details of our experimental settings.

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

## A    JUSTIFICATION FROM GENERALIZATION ERROR

We further analyze the generalization error bound of our `ProGrad`. We define the expected risk $\mathcal{R}(\cdot)$ and empirical risk $\hat{\mathcal{R}}(\cdot)$ of a classifier $f$ on domain $\mathcal{D}$ as

$$\mathcal{R}(f) = \mathbb{E}_{(X,Y) \sim \mathcal{D}}[\ell(f(X), Y)], \quad \hat{\mathcal{R}}(f) = \frac{1}{N} \sum_{i=1}^{N} \ell(f(X_i), Y_i) \tag{9}$$

where $\ell(f(X), Y)$ denotes the cross-entropy and $N$ is the volume of training data. We are interested in the downstream domain $\mathcal{D}_d$ and pre-trained domain $\mathcal{D}_p$, respectively. [1]

Let $\mathcal{F}$ be a function class, the conventional fine-tune model $\hat{f}_{\text{coop}}$ is trained on $\mathcal{D}_d$ by

$$\hat{f}_{\text{coop}} = \operatorname*{argmin}_{f \in \mathcal{F}} \hat{\mathcal{R}}_d(f). \tag{10}$$

The zero-shot CLIP model $\hat{f}_{\text{p}}$ is considered to be trained on $\mathcal{D}_p$ by

$$\hat{f}_{\text{p}} = \operatorname*{argmin}_{f \in \mathcal{F}} \hat{\mathcal{R}}_p(f). \tag{11}$$

For the implementation of `ProGrad`, we initialize the model $\hat{f}_{\text{prograd}}$ using the pre-trained model $\hat{f}_{\text{p}}$. We regularize each training step not to increase the KL divergence between the predictions of $\hat{f}_{\text{prograd}}$ and $\hat{f}_{\text{p}}$. In this way, $\hat{f}_{\text{prograd}}$ can keep the optimal value of the pre-trained domain $\mathcal{L}_{\text{kl}}$ when optimizing the empirical risk on the downstream domain. The model $\hat{f}_{\text{prograd}}$ learned by our `ProGrad` can be viewed as optimizing the empirical risk on both domains:

$$\hat{f}_{\text{prograd}} = \operatorname*{argmin}_{f \in \mathcal{F}} \hat{\mathcal{R}}_{(d+p)}(f) = \operatorname*{argmin}_{f \in \mathcal{F}} \hat{\mathcal{R}}_d(f) + \hat{\mathcal{R}}_p(f). \tag{12}$$

Based on Theorem 4.1 of (Yang et al., 2021), assuming that the neural network has $L$ layers with parameters matrices $W_1, ..., W_L$, and their Frobenius norm are at most $M_1, ..., M_L$ and the activation functions are 1-Lispschitz continuous, positive-homogeneous, and applied element-wise. The output of the neural network is the softmax function that predicts $c$ classes. Let $\mathcal{F}$ be a function class with the range $[a, b]$. Distribution is such that $\|\mathbf{x}\| \leq B$. Let $\mathbf{X}_1^{N_d} = \{\mathbf{x}_n^{(d)}\}_{n=1}^{N_d}$ and $\mathbf{X}_1^{N_p} = \{\mathbf{x}_n^{(p)}\}_{n=1}^{N_p}$ be two set of i.i.d. samples drawn from the downstream domain $\mathcal{D}_d$ and the pre-trained domain $\mathcal{D}_p$. Then for any $\epsilon > 0$, we have with probability at least $1 - \epsilon$,

---

[1] The pre-trained dataset includes samples from diverse classes. Here, we only consider the pre-trained data belonging to the classes of downstream task.

$$\mathcal{R}_d(\hat{f}_{\text{prograd}}) \leq \hat{\mathcal{R}}_{(d+p)}(\hat{f}_{\text{prograd}}) + \frac{1}{2}\gamma_{\mathcal{F}}(D,P) + \frac{cB\left(\sqrt{2\log(2)L}+1\right)\prod_{j=1}^{L}M_j}{\sqrt{N_p}}$$

$$+ \frac{cB\left(\sqrt{2\log(2)L}+1\right)\prod_{j=1}^{L}M_j}{\sqrt{N_d}} + \frac{3}{2}\sqrt{\frac{(b-a)\ln(4/\epsilon)}{2N_d}} \tag{13}$$

$$+ \frac{3}{2}\sqrt{\frac{(b-a)\ln(4/\epsilon)}{2N_p}} + \frac{1}{2}\sqrt{\frac{(b-a)^2\ln(4/\epsilon)}{2}\left(\frac{1}{N_d}+\frac{1}{N_p}\right)},$$

where $\gamma_{\mathcal{F}}(D,P)$ is the integral probability metric (Müller, 1997) that measures the difference between the distribution of pre-trained domain and the downstream domain. The Eq. (13) shows that the generalization error $\mathcal{R}_d(\hat{f}_{\text{prograd}})$ is bounded by the empirical training risk $\hat{\mathcal{R}}_{(d+p)}(\hat{f}_{\text{prograd}})$, the two domain gap $\gamma_{\mathcal{F}}(D,P)$ and the estimation error that is inversely proportional to number of training samples, *i.e.*, $N_d$ and $N_p$. The empirical training risk can be minimized to arbitrary small value and the estimation error that related to $N_p$ asymptotically tends to 0 as the sample size $N_p$ tends to infinity. Thanks to the large amount of pretrained samples $N_p$, we can approximate the generalization error bound for the model learned by `ProGrad` as

$$\mathcal{R}_d(\hat{f}_{\text{prograd}}) \leq \frac{1}{2}\gamma_{\mathcal{F}}(S,P) + \frac{cB\left(\sqrt{2\log(2)L}+1\right)\prod_{j=1}^{L}M_j}{\sqrt{N_d}}$$

$$+ \frac{3}{2}\sqrt{\frac{(b-a)\ln(4/\epsilon)}{2N_d}} + \frac{1}{2}\sqrt{\frac{(b-a)^2\ln(4/\epsilon)}{2}\frac{1}{N_d}}. \tag{14}$$

Similarly, we have the generalization error for $\hat{f}_{\text{coop}}$ as

$$\mathcal{R}_d(\hat{f}_{\text{coop}}) \leq 2\frac{cB\left(\sqrt{2\log(2)L}+1\right)\prod_{j=1}^{L}M_j}{\sqrt{N_d}} + 3\sqrt{\frac{(b-a)\ln(4/\epsilon)}{2N_d}} + \sqrt{\frac{(b-a)^2\ln(4/\epsilon)}{2}\frac{1}{N_d}}. \tag{15}$$

If the gap between the pre-trained domain $\mathcal{D}_p$ and the downstream domain $\mathcal{D}_d$ is very small, the $\gamma_{\mathcal{F}}(D,P)$ will tend to 0. Under this assumption, the estimation error bound of $\mathcal{R}_d(\hat{f}_{\text{coop}})$ is at least 2 times greater than $\mathcal{R}_d(\hat{f}_{\text{prograd}})$. Considering that in few-shot setting, $N_d$ is typically very small, which makes our `ProGrad` model $\hat{f}_{\text{prograd}}$ a much lower error bound than conventional fine-tuning model $\hat{f}_{\text{coop}}$.

## B   ADDITIONAL IMPLEMENTATION DETAILS

For `ProGrad` implementation, we first initialize the learnable context vector $v$ with the word embeddings of the zero-shot hand-crafted prompt. Concretely, if the context length $M$ is 16 and the hand-crafted prompt is "`a photo of a`", which only has 4 tokens, we initialize the former 12 context vectors with zeros and the last 4 context vectors with the word embedding of "`a photo of a`". We follow the training settings of CoOp (Zhou et al., 2021): All prompt-based models are trained by SGD with an initial learning rate of 0.002 which is decayed by the cosine annealing rule. During the first epoch, we use the warm-up trick by fixing the learning rate to $1 \times 10^{-5}$ to alleviate the gradient explosion. The training epoch is set to 50 for all shots of experiments of ImageNet dataset. For the rest 10 datasets, the training epoch is set to 50 for 1 shot, 100 for 2/4 shots and 200 for 8/16 shots. We train all prompt-based model with batch size of 32 expect for CoCoOp. As described in (Zhou et al., 2022), CoCoOp consumes a significant amount of GPU memory if the batch size is set larger than one. We set the batch size to 1, following their original setting. Our experiments are conducted on one 2080Ti GPU for all datasets except ImageNet where we train the models on one A100 GPU.

Table 6: Hand-crafted Prompts.

| Dataset | Hand-crafted prompt |
|---------|---------------------|
| OxfordPets | `"a type of pet, a photo of a {}."` |
| OxfordFlowers | `"a type of flower, a photo of a {}."` |
| FGVCAircraft | `"a type of aircraft, a photo of a {}.` |
| DescribableTextures | `"a texture of {}."` |
| EuroSAT | `"a centered satellite photo of {}."` |
| StanfordCars | `"a photo of a {}."` |
| Food101 | `"a type of food, a photo of {}."` |
| SUN397 | `"a photo of a {}."` |
| Caltech101 | `"a photo of a {}."` |
| UCF101 | `"a photo of a person doing {}."` |
| ImageNet | `"a photo of a {}."` |
| ImageNetSketch | `"a photo of a {}."` |
| ImageNetV2 | `"a photo of a {}."` |
| ImageNetA | `"a photo of a {}."` |
| ImageNetR | `"a photo of a {}."` |

Table 7: Prompt Ensembling Examples for ImageNet.

```
"a bad photo of a {}." "a photo of many {}." "a sculpture of a {}."
"a photo of the hard to see {}." "a low resolution photo of the {}."
"a rendering of a {}." "graffiti of a {}." "a bad photo of the {}."
"a cropped photo of the {}." "a tattoo of a {}." "the embroidered {}."
"a photo of a hard to see {}." "a bright photo of a {}."
"a photo of a clean {}." "a photo of a dirty {}."
"a dark photo of the {}." "a drawing of a {}."
"a photo of my {}." "the plastic {}." "a photo of the cool {}."
"a close-up photo of a {}." "a black and white photo of the {}."
"a painting of the {}." "a painting of a {}."
"a pixelated photo of the {}." "a sculpture of the {}."
"a bright photo of the {}." "a cropped photo of a {}." "a plastic {}."
"a photo of the dirty {}." "a jpeg corrupted photo of a {}."
"a blurry photo of the {}." "a photo of the {}." "a good photo of the {}."
"a rendering of the {}." "a {} in a video game.' "a photo of one {}."
"a doodle of a {}." "a close-up photo of the {}." "a photo of a {}."
"the origami {}." "the {} in a video game.' "a sketch of a {}."
"a doodle of the {}." "a origami {}." "a low resolution photo of a {}."
"the toy {}." "a rendition of the {}." "a photo of the clean {}."
"a photo of a large {}." "a rendition of a {}." "a photo of a nice {}."
"a photo of a weird {}." "a blurry photo of a {}." "a cartoon {}."
"art of a {}." "a sketch of the {}." "a embroidered {}."
"a pixelated photo of a {}." "itap of the {}."
"a jpeg corrupted photo of the {}." "a good photo of a {}."
"a plushie {}." "a photo of the nice {}." "a photo of the small {}."
"a photo of the weird {}." "the cartoon {}." "art of the {}."
"a drawing of the {}." "a photo of the large {}."
"a black and white photo of a {}." "the plushie {}."
"a dark photo of a {}." "itap of a {}."
"graffiti of the {}." "a toy {}." "itap of my {}."
"a photo of a cool {}." "a photo of a small {}." "a tattoo of the {}."
```

## C  HAND-CRAFTED PROMPTS

The hand-crafted prompts for 11 datasets as well as the ImageNet variants are listed in Table 6. We select the ensemble prompts from CLIP (Radford et al., 2021), examples for ImageNet are shown in Table 7.

Table 8: **Comparison of prompt learning methods in the cross-dataset transfer setting**. Prompts are learned from 4-shots ImageNet.

| | Source | Target | | | | | | | | | | |
|---|---|---|---|---|---|---|---|---|---|---|---|---|
| | ImageNet | Caltech101 | OxfordPets | StanfordCars | Flowers102 | Food101 | FGVCAircraft | SUN397 | DTD | EuroSAT | UCF101 | Average |
| CoOp | 61.34 | 84.48 | 85.99 | 54.16 | 60.10 | 75.48 | 14.09 | 57.48 | 35.32 | **26.72** | 57.56 | 55.70 |
| CoCoOp | 61.04 | 84.73 | 86.42 | 52.34 | 61.24 | 73.79 | 13.74 | 55.94 | 36.60 | 23.46 | 57.97 | 55.21 |
| ProGrad | **62.17** | **88.30** | **86.43** | **55.61** | **62.69** | **76.76** | **15.76** | **60.16** | **39.48** | 24.87 | **58.70** | **57.36** |

# D   ADDITIONAL EXPERIMENTS

## D.1   CROSS-DATASET TRANSFER

All models are trained on ImageNet as source dataset and evaluated on the rest 10 target datasets. The goal of this setting is to demonstrate the potential to transfer beyond a single dataset. The results are presented in Table 8. As shown, our `ProGrad` not only achieves the highest performance on source datasets but also outperforms other baselines 9 out of 10 datasets.

## D.2   UPPER BOUND OF PROGRAD

As the regularized general gradient direction $G_g$ is the key to improve the results, we are interested in the upper-bound of performance if we use an oracle direction $G_g^{full}$ instead of the one offered by hand-crafted prompt. To do so, we first optimize a prompt with plain cross-entropy loss on the full dataset to create $G_g^{full}$ and then use such gradient to implement ProGrad. The results are shown in Table 9. The results indicate a more accurate regularization direction $G_g^{full}$ can elicit a stronger ProGrad model.

Table 9: Upper-bound results (%) of 1, 2, 4, 8, and 16 shots of few-shot learning.

(a) Caltech101.

| #shots | 1 | 2 | 4 | 8 | 16 |
|---|---|---|---|---|---|
| ProGrad | 88.68 | 87.98 | 89.99 | 89.99 | 92.17 |
| ProGrad$_{full}$ | 91.08 | 91.70 | 91.76 | 91.84 | 92.86 |

(b) DTD.

| #shots | 1 | 2 | 4 | 8 | 16 |
|---|---|---|---|---|---|
| ProGrad | 46.14 | 49.78 | 54.43 | 60.69 | 63.97 |
| ProGrad$_{full}$ | 66.11 | 67.04 | 67.24 | 68.46 | 69.27 |

(c) Stanford Cars.

| #shots | 1 | 2 | 4 | 8 | 16 |
|---|---|---|---|---|---|
| ProGrad | 58.38 | 61.81 | 65.62 | 69.29 | 73.75 |
| ProGrad$_{full}$ | 70.54 | 71.57 | 71.66 | 72.73 | 75.27 |

## D.3   EFFECT OF HYPER-PARAMETER

We further analyze the effect of the hyper-parameter $\lambda$ described in Eq. (4) in the main paper. Results are shown in Table. 10. As discussed in Section 3.2 in the main paper, a smaller $\lambda$ weakens the general knowledge regularization, which results in a inferior performance under low-shot setting for most datasets. However, for DTD in Table 10, using a smaller $\lambda = 0.9$ to reduce the general knowledge

regularization can improve the 16 shots results. One possible reason is that texture images of DTD has large gap with the CLIP pre-trained images that collected from the Internet, stronger regularization from pre-trained knowledge might be detrimental to the fine-tune performance if downstream data is sufficient.

Table 10: Accuracy (%) of 1, 2, 4, 8, and 16 shots training with different $\lambda$ on DTD and OxfordPets.

| (a) **OxfordPets**. | | | | | | (b) **DTD**. | | | | |
|---|---|---|---|---|---|---|---|---|---|---|
| $\lambda$ | 1 shot | 2 shots | 4 shots | 8 shots | 16 shots | $\lambda$ | 1 shot | 2 shots | 4 shots | 8 shots | 16 shots |
| 0 | 86.01 | 82.21 | 86.63 | 85.15 | 87.06 | 0 | 44.33 | 46.22 | 54.08 | 59.99 | 62.89 |
| 0.2 | 87.12 | 83.16 | 84.87 | 84.00 | 86.15 | 0.2 | 43.80 | 45.21 | 54.02 | 60.14 | 63.61 |
| 0.4 | 88.09 | 83.56 | 85.50 | 84.04 | 86.67 | 0.4 | 44.17 | 47.44 | 54.32 | 59.65 | 63.16 |
| 0.7 | 87.74 | 84.70 | 86.93 | 86.30 | 87.90 | 0.7 | 44.93 | 47.77 | 54.92 | 59.28 | 63.10 |
| 0.9 | 88.26 | 86.47 | 87.52 | 87.38 | 88.52 | 0.9 | 45.78 | 48.46 | 55.44 | 60.46 | 64.28 |
| 1.0 | 88.36 | 86.89 | 88.04 | 87.91 | 89.00 | 1.0 | 46.14 | 49.78 | 54.43 | 57.98 | 61.15 |

## D.4    ADDITIONAL FEW-SHOT CLASSIFICATION RESULTS

In this section, we further provide the detailed few-shot classification results of other learning-based fine-tuning methods with confidence interval at 95% in Table 11 and Table 12.

**Cosine.** As described in Section 4.5 of the main paper, we plug in an additional cosine classifier on top of the visual backbone and trained on downstream dataset.

**CoOp** learns the context prompt from data rather than hand-crafted design.

**CLIP-Adapter** learns additional feature adapter to boost conventional fine-tuning results.

**Cosine + ProGrad** employs `ProGrad` to the training process of cosine classifier.

**CoOp + $l_2$ prompt reg**. We further investigate whether simply using the $l_2$ distance between learned prompt vector $v$ and the word embedding vector of hand-crafted prompt $v_{zs}$ as the regularization can improve few-shot performance, *i.e.*, $\mathcal{L}_{\text{total}}(v) = \mathcal{L}_{\text{ce}}(v) + \alpha\|v - v_{zs}\|_2$, where we select $\alpha = 0.01$.

**CoOp + GM** applies gradient matching method (Yu et al., 2020) to CoOp, *i.e.*, we not only project the $G_d$ to the perpendicular direction of $G_g$ as the updated gradient, but also project the $G_g$ to the perpendicular direction of $G_d$ as the updated gradient to fine-tune the model alternately.

**CoOp + KD.** As described in Section 4.5 of the main paper, we apply knowledge distillation loss to CoOp, *i.e.*, $\mathcal{L}_{\text{total}} = \mathcal{L}_{\text{ce}} + \mathcal{L}_{\text{kl}}$

**CoOp + ProGrad** employs `ProGrad` to CoOp.

For all prompt-based methods, we set the context length $M$ to 16 except for CoOp + $l_2$ prompt reg. The learned length for CoOp + $l_2$ prompt reg needs to be equal to the hand-crafted prompt length to compute the $l_2$ norm, *e.g.*, the M has to be 4 if the hand-crafted prompt is "`a photo of a` ". According to the average results in Table 11, we observe that our CoOp + ProGrad still achieves the best average performance. By comparing the results of 1) Cosine and Cosine + ProGrad; and 2) CoOp and CoOp + ProGrad, we demonstrates both conventional "pre-train then fine-tune" paradigm and prompt tuning paradigm can benefit from our `ProGrad`. The gap between CoOp and CoOp + $l_2$ prompt reg demonstrates that directly regularize the learned prompt to be not far away from the hand-crafted prompt has limited improvement. By digging into CoOp + KD and CoOp + GM, we find performance improvement by introducing the general knowledge. However, their performance still under-performs our CoOp + ProGrad. This is because 1) CoOp + KD learns the average knowledge from two domains which still allows the fine-tuned model to learn from the downstream knowledge that conflicts with the general knowledge; 2) CoOp + MD additional requires the fine-tuned model to discards the general knowledge that is not aligned with the downstream knowledge, as the downstream data is limited, the inaccurate estimation of $G_d$ will lead the model focus on biased general knowledge.

## D.5    ADDITIONAL RESULTS FOR BASE-TO-NEW GENERALIZATION

Table 13 further presents the results for base-to-new generalization on each of the 11 datasets.

Table 11: Accuracy (%) with confidence interval at 95% of few-shot learning on 11 datasets (**Part I**). The context length $M$ is set 16 for prompt-based methods. * indicates results copied from (Gao et al., 2021a).

| | Method | #shots per class | | | | |
|---|---|---|---|---|---|---|
| | | 1 | 2 | 4 | 8 | 16 |
| Average | Cosine | $30.50 \pm 1.24$ | $43.74 \pm 1.37$ | $53.33 \pm 1.57$ | $61.26 \pm 1.45$ | $65.00 \pm 2.87$ |
| | CoOp | $59.44 \pm 1.88$ | $62.31 \pm 1.40$ | $66.72 \pm 0.93$ | $70.06 \pm 0.53$ | $73.48 \pm 0.39$ |
| | CLIP-Adapter* | $61.45$ | $64.32$ | $67.51$ | $70.78$ | $74.35$ |
| | Cosine + ProGrad | $32.29 \pm 1.12$ | $46.14 \pm 1.49$ | $55.18 \pm 1.99$ | $62.05 \pm 0.93$ | $66.47 \pm 1.69$ |
| | CoOp + $l_2$ prompt reg | $60.84 \pm 1.16$ | $62.75 \pm 1.18$ | $66.85 \pm 0.76$ | $70.08 \pm 0.58$ | $72.92 \pm 0.46$ |
| | CoOp + GM | $61.27 \pm 0.96$ | $63.23 \pm 0.50$ | $64.59 \pm 0.63$ | $66.40 \pm 0.49$ | $67.12 \pm 0.29$ |
| | CoOp + KD | $61.52 \pm 0.99$ | $64.07 \pm 0.52$ | $66.52 \pm 0.38$ | $70.01 \pm 0.31$ | $72.01 \pm 0.37$ |
| | CoOp + ProGrad | $62.61 \pm 0.80$ | $64.90 \pm 0.86$ | $68.45 \pm 0.52$ | $71.41 \pm 0.49$ | $74.28 \pm 0.40$ |
| ImageNet | Cosine | $15.95 \pm 0.07$ | $26.56 \pm 0.30$ | $37.08 \pm 0.29$ | $46.18 \pm 0.19$ | $53.36 \pm 0.39$ |
| | CoOp | $57.15 \pm 1.03$ | $57.25 \pm 0.43$ | $59.51 \pm 0.25$ | $61.59 \pm 0.17$ | $63.00 \pm 0.18$ |
| | CLIP-Adapter* | $58.14$ | $58.55$ | $59.41$ | $60.36$ | $61.27$ |
| | Cosine + ProGrad | $19.21 \pm 0.28$ | $31.18 \pm 0.18$ | $42.59 \pm 0.29$ | $51.73 \pm 0.18$ | $57.65 \pm 0.33$ |
| | CoOp + $l_2$ prompt reg | $57.51 \pm 0.22$ | $61.27 \pm 0.49$ | $62.49 \pm 0.12$ | $62.71 \pm 0.01$ | $62.88 \pm 0.09$ |
| | CoOp + GM | $60.41 \pm 0.17$ | $60.51 \pm 0.13$ | $60.75 \pm 0.06$ | $61.01 \pm 0.14$ | $61.44 \pm 0.03$ |
| | CoOp + KD | $60.85 \pm 0.22$ | $61.08 \pm 0.10$ | $61.51 \pm 0.07$ | $61.67 \pm 0.12$ | $62.05 \pm 0.09$ |
| | CoOp + ProGrad | $57.75 \pm 0.24$ | $59.75 \pm 0.33$ | $61.46 \pm 0.07$ | $62.54 \pm 0.03$ | $63.45 \pm 0.08$ |
| Caltech101 | Cosine | $60.76 \pm 1.71$ | $73.10 \pm 1.01$ | $81.43 \pm 0.65$ | $87.02 \pm 0.60$ | $90.60 \pm 0.05$ |
| | CoOp | $87.40 \pm 0.98$ | $87.92 \pm 1.12$ | $89.48 \pm 0.47$ | $90.25 \pm 0.18$ | $92.00 \pm 0.02$ |
| | CLIP-Adapter* | $88.52$ | $89.19$ | $91.04$ | $91.71$ | $93.42$ |
| | Cosine + ProGrad | $61.95 \pm 0.12$ | $75.24 \pm 0.88$ | $82.98 \pm 0.38$ | $88.59 \pm 0.21$ | $91.31 \pm 0.19$ |
| | CoOp + $l_2$ prompt reg | $87.04 \pm 0.61$ | $87.37 \pm 0.78$ | $88.82 \pm 0.40$ | $89.62 \pm 0.29$ | $91.67 \pm 0.26$ |
| | CoOp + GM | $89.14 \pm 0.15$ | $89.37 \pm 0.26$ | $89.64 \pm 0.33$ | $89.36 \pm 0.31$ | $89.42 \pm 0.13$ |
| | CoOp + KD | $89.06 \pm 0.29$ | $89.71 \pm 0.20$ | $90.13 \pm 0.16$ | $90.09 \pm 0.30$ | $91.39 \pm 0.05$ |
| | CoOp + ProGrad | $88.68 \pm 0.34$ | $87.98 \pm 0.69$ | $89.99 \pm 0.26$ | $90.83 \pm 0.07$ | $92.17 \pm 0.17$ |
| OxfordPets | Cosine | $26.33 \pm 0.75$ | $41.60 \pm 1.93$ | $55.29 \pm 1.97$ | $66.60 \pm 0.82$ | $66.84 \pm 16.24$ |
| | CoOp | $86.01 \pm 0.47$ | $82.21 \pm 2.12$ | $86.63 \pm 1.02$ | $85.15 \pm 1.12$ | $87.06 \pm 0.88$ |
| | CLIP-Adapter* | $81.44$ | $81.57$ | $82.69$ | $84.13$ | $85.31$ |
| | Cosine + ProGrad | $26.08 \pm 0.73$ | $40.58 \pm 2.01$ | $55.23 \pm 1.44$ | $66.78 \pm 1.58$ | $68.96 \pm 14.35$ |
| | CoOp + $l_2$ prompt reg | $87.55 \pm 0.15$ | $82.12 \pm 2.61$ | $84.93 \pm 1.77$ | $84.38 \pm 0.75$ | $86.28 \pm 0.45$ |
| | CoOp + GM | $87.05 \pm 0.65$ | $87.06 \pm 0.67$ | $88.45 \pm 0.45$ | $88.35 \pm 0.15$ | $88.38 \pm 0.27$ |
| | CoOp + KD | $87.10 \pm 1.47$ | $87.40 \pm 0.60$ | $88.56 \pm 0.19$ | $88.77 \pm 0.24$ | $89.16 \pm 0.16$ |
| | CoOp + ProGrad | $88.36 \pm 0.73$ | $86.89 \pm 0.42$ | $88.04 \pm 0.50$ | $87.91 \pm 0.54$ | $89.00 \pm 0.32$ |
| StanfordCars | Cosine | $18.96 \pm 0.34$ | $33.37 \pm 0.38$ | $47.75 \pm 0.38$ | $61.30 \pm 0.25$ | $71.94 \pm 0.31$ |
| | CoOp | $55.68 \pm 1.23$ | $58.33 \pm 0.60$ | $63.05 \pm 0.09$ | $68.37 \pm 0.25$ | $73.34 \pm 0.49$ |
| | CLIP-Adapter* | $56.02$ | $58.24$ | $63.07$ | $67.00$ | $72.83$ |
| | Cosine + ProGrad | $21.13 \pm 0.50$ | $39.44 \pm 0.83$ | $54.54 \pm 0.57$ | $66.47 \pm 0.14$ | $73.41 \pm 0.11$ |
| | CoOp + $l_2$ prompt reg | $55.86 \pm 0.66$ | $57.69 \pm 0.51$ | $62.82 \pm 0.07$ | $66.63 \pm 0.25$ | $69.86 \pm 0.44$ |
| | CoOp + GM | $57.37 \pm 0.36$ | $58.46 \pm 0.24$ | $59.72 \pm 0.66$ | $62.32 \pm 0.59$ | $63.87 \pm 0.37$ |
| | CoOp + KD | $57.48 \pm 1.47$ | $59.09 \pm 0.60$ | $61.47 \pm 0.19$ | $67.73 \pm 0.24$ | $70.48 \pm 0.16$ |
| | CoOp + ProGrad | $58.38 \pm 0.23$ | $61.81 \pm 0.45$ | $65.62 \pm 0.43$ | $69.29 \pm 0.11$ | $73.46 \pm 0.29$ |
| Flowers102 | Cosine | $51.33 \pm 2.77$ | $70.06 \pm 2.29$ | $82.43 \pm 1.65$ | $91.74 \pm 0.73$ | $95.68 \pm 0.22$ |
| | CoOp | $68.13 \pm 1.74$ | $76.68 \pm 1.82$ | $86.13 \pm 0.75$ | $91.74 \pm 0.49$ | $94.72 \pm 0.34$ |
| | CLIP-Adapter* | $71.97$ | $78.80$ | $85.31$ | $90.69$ | $94.30$ |
| | Cosine + ProGrad | $52.08 \pm 2.31$ | $70.13 \pm 1.90$ | $81.09 \pm 2.06$ | $91.62 \pm 0.41$ | $93.94 \pm 0.02$ |
| | CoOp + $l_2$ prompt reg | $71.12 \pm 0.55$ | $80.36 \pm 0.54$ | $86.42 \pm 0.33$ | $91.58 \pm 0.59$ | $94.25 \pm 0.38$ |
| | CoOp + GM | $67.87 \pm 0.31$ | $69.09 \pm 0.49$ | $71.69 \pm 0.68$ | $75.76 \pm 0.79$ | $78.36 \pm 0.34$ |
| | CoOp + KD | $68.11 \pm 1.47$ | $71.02 \pm 0.60$ | $76.06 \pm 0.19$ | $84.53 \pm 0.24$ | $88.05 \pm 0.16$ |
| | CoOp + ProGrad | $73.18 \pm 0.73$ | $79.77 \pm 0.65$ | $85.37 \pm 0.96$ | $91.64 \pm 0.24$ | $94.37 \pm 0.24$ |

Table 12: Accuracy (%) with confidence interval at 95% of few-shot learning on 11 datasets (**Part II**). The context length $M$ is set 16 for prompt-based methods. * indicates results copied from (Gao et al., 2021a).

| | Method | #shots per class | | | | |
|---|---|---|---|---|---|---|
| | | 1 | 2 | 4 | 8 | 16 |
| Food101 | Cosine | $25.32 \pm 0.29$ | $41.06 \pm 0.29$ | $54.10 \pm 1.06$ | $61.88 \pm 0.33$ | $68.50 \pm 0.24$ |
| | CoOp | $74.28 \pm 1.40$ | $72.45 \pm 1.29$ | $73.27 \pm 2.07$ | $71.67 \pm 0.30$ | $74.68 \pm 0.03$ |
| | CLIP-Adapter* | 75.09 | 75.59 | 75.92 | 76.53 | 76.97 |
| | Cosine + ProGrad | $27.19 \pm 0.15$ | $45.28 \pm 0.36$ | $58.57 \pm 1.01$ | $71.25 \pm 0.29$ | $75.61 \pm 0.15$ |
| | CoOp + $l_2$ prompt reg | $73.58 \pm 2.20$ | $68.89 \pm 1.30$ | $71.30 \pm 0.49$ | $72.42 \pm 0.26$ | $75.64 \pm 0.33$ |
| | CoOp + GM | $76.23 \pm 1.51$ | $77.97 \pm 0.51$ | $78.89 \pm 0.10$ | $78.90 \pm 0.15$ | $79.07 \pm 0.06$ |
| | CoOp + KD | $76.06 \pm 1.47$ | $77.59 \pm 0.60$ | $78.72 \pm 0.19$ | $78.38 \pm 0.24$ | $78.90 \pm 0.16$ |
| | CoOp + ProGrad | $76.04 \pm 0.54$ | $74.95 \pm 0.57$ | $75.95 \pm 0.27$ | $76.65 \pm 0.23$ | $78.41 \pm 0.08$ |
| FGVCAircraft | Cosine | $12.47 \pm 1.00$ | $17.75 \pm 1.35$ | $22.00 \pm 1.50$ | $29.14 \pm 0.54$ | $36.47 \pm 0.18$ |
| | CoOp | $9.71 \pm 6.09$ | $18.74 \pm 0.48$ | $21.78 \pm 0.50$ | $27.55 \pm 0.06$ | $31.37 \pm 0.53$ |
| | CLIP-Adapter* | 19.63 | 22.27 | 25.62 | 30.48 | 38.72 |
| | Cosine + ProGrad | $12.83 \pm 0.48$ | $17.59 \pm 1.59$ | $19.70 \pm 1.62$ | $26.34 \pm 0.51$ | $31.98 \pm 0.68$ |
| | CoOp + $l_2$ prompt reg | $18.01 \pm 0.44$ | $19.78 \pm 0.23$ | $22.51 \pm 0.94$ | $27.24 \pm 0.38$ | $30.55 \pm 0.54$ |
| | CoOp + GM | $17.08 \pm 0.37$ | $19.34 \pm 0.24$ | $19.62 \pm 0.40$ | $21.07 \pm 0.08$ | $22.52 \pm 0.19$ |
| | CoOp + KD | $17.67 \pm 0.45$ | $19.29 \pm 0.15$ | $21.21 \pm 0.60$ | $25.55 \pm 0.30$ | $28.58 \pm 0.42$ |
| | CoOp + ProGrad | $18.81 \pm 0.50$ | $20.47 \pm 0.90$ | $23.32 \pm 0.36$ | $27.02 \pm 0.67$ | $31.12 \pm 0.62$ |
| SUN397 | Cosine | $25.32 \pm 0.18$ | $38.13 \pm 0.37$ | $49.83 \pm 0.45$ | $56.97 \pm 0.21$ | $62.84 \pm 0.16$ |
| | CoOp | $60.30 \pm 0.64$ | $59.52 \pm 0.60$ | $63.33 \pm 0.39$ | $65.65 \pm 0.10$ | $69.14 \pm 0.11$ |
| | CLIP-Adapter* | 61.16 | 62.08 | 64.74 | 66.88 | 69.20 |
| | Cosine + ProGrad | $29.66 \pm 0.08$ | $45.81 \pm 0.39$ | $55.92 \pm 0.35$ | $63.61 \pm 0.16$ | $67.33 \pm 0.25$ |
| | CoOp + $l_2$ prompt reg | $57.64 \pm 0.33$ | $59.81 \pm 0.33$ | $64.88 \pm 0.45$ | $67.66 \pm 0.16$ | $69.56 \pm 0.11$ |
| | CoOp + GM | $62.73 \pm 0.35$ | $62.85 \pm 0.10$ | $63.32 \pm 0.21$ | $63.77 \pm 0.04$ | $64.47 \pm 0.27$ |
| | CoOp + KD | $62.89 \pm 0.40$ | $64.10 \pm 0.29$ | $65.83 \pm 0.26$ | $67.02 \pm 0.05$ | $68.32 \pm 0.19$ |
| | CoOp + ProGrad | $60.54 \pm 0.24$ | $63.06 \pm 0.11$ | $66.39 \pm 0.43$ | $67.62 \pm 0.28$ | $69.84 \pm 0.18$ |
| DTD | Cosine | $27.05 \pm 0.83$ | $38.42 \pm 0.48$ | $48.44 \pm 2.29$ | $58.47 \pm 0.51$ | $61.88 \pm 0.38$ |
| | CoOp | $43.77 \pm 2.12$ | $46.06 \pm 1.05$ | $53.82 \pm 0.77$ | $60.06 \pm 1.18$ | $63.26 \pm 0.22$ |
| | CLIP-Adapter* | 45.65 | 50.54 | 56.43 | 61.59 | 66.03 |
| | Cosine + ProGrad | $26.95 \pm 1.38$ | $38.87 \pm 1.02$ | $48.05 \pm 3.02$ | $56.24 \pm 2.81$ | $63.40 \pm 0.58$ |
| | CoOp + $l_2$ prompt reg | $43.74 \pm 1.45$ | $45.98 \pm 2.76$ | $53.25 \pm 1.55$ | $59.08 \pm 0.58$ | $62.31 \pm 1.05$ |
| | CoOp + GM | $43.81 \pm 2.15$ | $47.64 \pm 0.63$ | $49.17 \pm 1.52$ | $53.17 \pm 0.63$ | $54.06 \pm 0.45$ |
| | CoOp + KD | $43.01 \pm 2.18$ | $49.31 \pm 1.10$ | $53.03 \pm 1.49$ | $60.26 \pm 0.34$ | $63.14 \pm 0.39$ |
| | CoOp + ProGrad | $46.14 \pm 1.74$ | $49.78 \pm 1.37$ | $54.43 \pm 0.86$ | $60.69 \pm 0.10$ | $63.97 \pm 0.61$ |
| EuroSAT | Cosine | $37.55 \pm 5.27$ | $52.93 \pm 5.66$ | $49.81 \pm 6.23$ | $46.08 \pm 11.13$ | $33.30 \pm 13.04$ |
| | CoOp | $49.40 \pm 3.86$ | $62.23 \pm 4.94$ | $69.49 \pm 3.23$ | $76.56 \pm 1.73$ | $84.05 \pm 1.05$ |
| | CLIP-Adapter* | 54.53 | 63.73 | 68.33 | 75.81 | 82.81 |
| | Cosine + ProGrad | $41.55 \pm 6.19$ | $51.35 \pm 5.76$ | $47.64 \pm 9.68$ | $30.03 \pm 2.99$ | $33.30 \pm 1.67$ |
| | CoOp + $l_2$ prompt reg | $54.28 \pm 5.38$ | $62.60 \pm 2.77$ | $70.43 \pm 1.81$ | $77.32 \pm 2.20$ | $83.30 \pm 1.11$ |
| | CoOp + GM | $48.02 \pm 4.04$ | $57.12 \pm 2.03$ | $62.88 \pm 2.28$ | $68.74 \pm 2.18$ | $67.72 \pm 1.00$ |
| | CoOp + KD | $49.51 \pm 1.12$ | $58.89 \pm 1.06$ | $66.79 \pm 0.76$ | $74.37 \pm 0.91$ | $77.87 \pm 1.74$ |
| | CoOp + ProGrad | $56.32 \pm 3.04$ | $63.10 \pm 3.77$ | $72.53 \pm 1.29$ | $78.04 \pm 2.45$ | $83.74 \pm 0.70$ |
| UCF101 | Cosine | $34.41 \pm 0.40$ | $48.21 \pm 1.00$ | $58.47 \pm 0.81$ | $68.46 \pm 0.66$ | $73.64 \pm 0.32$ |
| | CoOp | $62.03 \pm 1.13$ | $63.98 \pm 0.91$ | $67.45 \pm 0.74$ | $72.11 \pm 0.29$ | $75.67 \pm 0.49$ |
| | CLIP-Adapter* | 63.80 | 66.98 | 70.07 | 73.45 | 76.99 |
| | Cosine + ProGrad | $36.61 \pm 0.14$ | $52.11 \pm 1.43$ | $60.66 \pm 1.50$ | $69.85 \pm 0.94$ | $74.27 \pm 0.30$ |
| | CoOp + $l_2$ prompt reg | $62.88 \pm 0.74$ | $64.43 \pm 0.71$ | $67.46 \pm 0.40$ | $72.28 \pm 0.88$ | $75.77 \pm 0.29$ |
| | CoOp + GM | $64.27 \pm 0.48$ | $66.14 \pm 0.25$ | $66.37 \pm 0.27$ | $67.91 \pm 0.29$ | $68.96 \pm 0.04$ |
| | CoOp + KD | $64.99 \pm 0.35$ | $67.29 \pm 0.46$ | $68.44 \pm 0.13$ | $71.77 \pm 0.41$ | $74.15 \pm 0.55$ |
| | CoOp + ProGrad | $64.55 \pm 0.50$ | $66.35 \pm 0.18$ | $69.86 \pm 0.30$ | $73.33 \pm 0.65$ | $77.28 \pm 0.96$ |

Table 13: Accuracy (%) for the base-to-new generalization evaluation. The context length $M$ is 4 for prompt-based methods which are learned from the base classes with 4 shots. H: Harmonic mean.

| (a) **Average over 11 datasets**. | Base | New | H |
|---|---|---|---|
| CLIP | 61.72 | 65.91 | 63.64 |
| CoOp | 71.96 | 61.26 | 65.58 |
| CoCoOp | 72.23 | 60.77 | 65.35 |
| ProGrad | **73.29** | **65.96** | **69.06** |

| (b) ImageNet. | Base | New | H |
|---|---|---|---|
| CLIP | 64.46 | 59.99 | 62.14 |
| CoOp | 65.49 | 57.70 | 61.35 |
| CoCoOp | 66.21 | 58.01 | 61.84 |
| ProGrad | **66.96** | **60.04** | **63.23** |

| (c) Caltech101. | Base | New | H |
|---|---|---|---|
| CLIP | 90.90 | 90.72 | 90.81 |
| CoOp | 94.38 | 87.48 | 90.80 |
| CoCoOp | 94.43 | 87.81 | 91.00 |
| ProGrad | **94.47** | **90.84** | **92.46** |

| (d) OxfordPets. | Base | New | H |
|---|---|---|---|
| CLIP | 85.86 | 93.85 | 89.68 |
| CoOp | 90.31 | 94.03 | 92.13 |
| CoCoOp | 89.07 | 91.00 | 90.02 |
| ProGrad | **91.78** | **94.86** | **93.29** |

| (e) StanfordCars. | Base | New | H |
|---|---|---|---|
| CLIP | 55.55 | **66.35** | 60.47 |
| CoOp | 61.77 | 62.51 | 62.14 |
| CoCoOp | 61.68 | 59.98 | 60.82 |
| ProGrad | **63.01** | 64.32 | **63.66** |

| (f) Flowers102. | Base | New | H |
|---|---|---|---|
| CLIP | 64.10 | **70.92** | 67.34 |
| CoOp | **89.33** | 62.77 | 73.73 |
| CoCoOp | 88.07 | 66.26 | 75.62 |
| ProGrad | 88.19 | 69.38 | **77.66** |

| (g) Food101. | Base | New | H |
|---|---|---|---|
| CLIP | 81.48 | 82.15 | 81.81 |
| CoOp | 80.40 | 81.09 | 80.74 |
| CoCoOp | 79.77 | 77.68 | 78.71 |
| ProGrad | **83.10** | **83.57** | **83.33** |

| (h) FGVCAircraft. | Base | New | H |
|---|---|---|---|
| CLIP | 17.89 | **25.13** | 20.90 |
| CoOp | 22.53 | 20.40 | 21.41 |
| CoCoOp | 22.73 | 19.40 | 20.93 |
| ProGrad | **22.77** | 24.24 | **23.48** |

| (i) SUN397. | Base | New | H |
|---|---|---|---|
| CLIP | 66.45 | **70.17** | 68.26 |
| CoOp | 71.48 | 65.57 | 68.40 |
| CoCoOp | 71.88 | 67.10 | 69.41 |
| ProGrad | **73.71** | 69.78 | **71.69** |

| (j) DTD. | Base | New | H |
|---|---|---|---|
| CLIP | 49.31 | **54.35** | 51.71 |
| CoOp | 67.71 | 43.92 | 53.28 |
| CoCoOp | 63.54 | 40.78 | 49.68 |
| ProGrad | **66.90** | 53.06 | **59.18** |

| (k) EuroSAT. | Base | New | H |
|---|---|---|---|
| CLIP | 39.26 | 43.62 | 41.33 |
| CoOp | 73.53 | 40.19 | 51.97 |
| CoCoOp | 83.63 | 40.95 | 54.98 |
| ProGrad | 79.67 | **49.99** | **61.43** |

| (l) UCF101. | Base | New | H |
|---|---|---|---|
| CLIP | 63.70 | **67.71** | 65.64 |
| CoOp | 74.59 | 58.23 | 65.40 |
| CoCoOp | 73.51 | 59.55 | 65.80 |
| ProGrad | **75.66** | 65.52 | **70.23** |

