# OpenReview forum: "Prompt Tuning with Prompt-aligned Gradient for Vision-Language Models "
_ICLR.cc/2023/Conference — Submitted to ICLR 2023_

### Official Review · Reviewer_6JR9 · 2022-10-30

**Confidence:** 4
**Correctness:** 3
**Technical Novelty And Significance:** 3
**Empirical Novelty And Significance:** 3
**Recommendation:** 6

**Clarity, Quality, Novelty And Reproducibility:**

This paper is clear and of acceptable quality. The novelty might not be its main strength, but it is not a deal breaker either.

**Strength And Weaknesses:**

Strengths:

- The paper is based on a simple and effective idea and is executed well.
- The paper is accurately and clearly written.
- The performance is satisfactory.

Weaknesses :
- I see a lot of similarities between the proposed method with [1]. If I am not mistaken, the auxiliary loss function prevents forgetting the general knowledge is the same as [1]. However, this paper's gradient projection approach and theoretical analyses are new. While the authors limit their application to zero-shot learning, I believe the contributions of this paper hold in a general setting where some previous knowledge has to be preserved for training on a new task.

A discussion on the differences with [1] and direct comparisons with their method would be interesting and strengthen the paper, in my opinion. A quick experiment that can be done during the rebuttal is to show the performance of the proposed method without the gradient projection, i.e. adding the KL loss to CoOp and optimizing with the standard SGD. A similar experimental setup as [1] and director comparisons with them would be even more interesting.

[1] Zhizhong Li and Derek Hoiem, Learning without Forgetting, 2016

**Summary Of The Paper:**

This paper propses a method for prompt tuning that is less sensitive to overfitting compared to the CoOp approach.  The main idea is simple and effective, and is based on the observation that CoOp starts overftting when the training is not stoped early,  outpefroming Zero-shot CLIP. The authors propose to requalarize the CoOp objective via preserving the Zero-shot CLIP prediction, similar to learning without forgetting[1]. Moreover, they propse to modify the CoOp gradient in the direction of the regulariztion term gradient during the training. The proposed methods outperofrmrs not only CoOp approach but also more adnvced approaches like CoCoOp on the standrd zero-shot learning benchmarks.

[1] Zhizhong Li and Derek Hoiem, Learning without Forgetting, 2016

**Summary Of The Review:**

Overall, the idea of this paper is simple, effective, and executed well. More comparisons with [1] would be interesting to include. I'd be happy to discuss this matter during the rebuttal.

---

> ### Author Response · Authors · 2022-11-15
> **Response to reviewer 6JR9**
>
> **Q5.1** A discussion on the differences with ``Learning without Forgetting'' (LwF) and direction comparison with their method.
>
> **A5.1** Thanks for your constructive suggestions. The main difference between LwF and ProGrad is that ProGrad focuses on the downstream domain, i.e., in each optimization step, the updated gradient of ProGrad is identical to the downstream gradient offered by cross-entropy loss unless it conflicts with the pre-trained KL loss; on the other hand, LwF treats pre-trained and downstream domain as equally important, i.e., in each optimization step, LwF has to decrease both KL loss and cross-entropy loss. Because our target is to evaluate the performance on the downstream task. We expect the learned model to focus more on the downstream domain.
> Actually, in Table 4 of the main paper, we have already included the comparison by adding the KL loss to CoOp and optimizing with the standard SGD. We investigated whether LwF (i.e., the total loss is formulated as $\mathcal{L}_\text{total}=\mathcal{L}_\text{ce}+\alpha\cdot \mathcal{L}_\text{kl}$) can achieve similar performance as our ProGrad. We repeated the few-shot experiments on 11 datasets with a variety of $\alpha$ (from 0.25 to 4) and report the average results in Table 4 of the main paper. Overall, ProGrad outperforms LwF for various few-shot settings. Although LwF with small $\alpha \leq 1$ promotes CoOp in low-shot (e.g., 1, 2 and 4 shots), the performance drops when number of shots is large (see 8 and 16 shots). These results indicate that our ProGrad works differently from LwF and is more robust to the number of training samples.
> We further attach Table 4 as follows for your reference.
>
> | \#shots            | 1              | 2              | 4              | 8              | 16             |
> |--------------------|----------------|----------------|----------------|----------------|----------------|
> | CoOp               | 59.46          | 62.37          | 66.79          | 70.04          | 73.49          |
> | LwF, $\alpha=0.25$ | 61.09          | 63.01          | 67.74          | 70.90          | 73.39          |
> | LwF, $\alpha=0.5$  | 61.13          | 63.36          | 67.14          | 70.34          | 72.68          |
> | LwF, $\alpha=1$    | 61.52          | 64.07          | 66.52          | 70.01          | 72.01          |
> | LwF, $\alpha=2$    | 60.98          | 62.66          | 64.92          | 67.78          | 68.98          |
> | LwF, $\alpha=4$    | 59.58          | 61.76          | 62.92          | 65.01          | 65.42          |
> | ProGrad   | **62.61** |  **64.90** | **68.45** | **71.41** | **74.28** |
>
> *Title: Comparison with LwF. Average accuracy (\%) over 11 datasets.*

---

### Official Review · Reviewer_Wdn9 · 2022-10-30

**Confidence:** 4
**Correctness:** 4
**Technical Novelty And Significance:** 2
**Empirical Novelty And Significance:** 3
**Recommendation:** 6

**Clarity, Quality, Novelty And Reproducibility:**

Clarity: The paper is clearly written. All notations, concepts, ideas, experiments are properly defined or described.
Quality: This is an interesting work on an important research topic. Though there are still a few weaknesses, the overall quality is arguably above the average of all submissions that I have reviewed this year.
Novelty: The idea is novel, investigating of CLIP's few-shot generalization from an unexplored aspect.
Reproducibility: The computation trick is fully in Eq. 4 and one can easily re-implement the key experiments.

**Strength And Weaknesses:**

Strength:

- The work addressed some key challenging in prompt engineering for vision-language models.
- The idea is computationally simple and empirically validated to bring superior performance.

Weakness:

- The proof of generalization bound is not much informative. It is a direct extension of the theoretical results in prior works (e.g. Zhang and Ye's 2012 paper).
- More insights are required for the readers fully understanding the empirical superiority of the proposed idea. In specific,  the "general knowledge" induced by standard zero-shot CLIP transfer does not provide ground truth. The gradient-calculating rule as proposed in Eq. 4 is essentially a fusion of two kinds of cues: one is from zero-shot extension, and the other is data-driven. The trick is to cleverly choose the way for the fusion. In the practice of multi-source fusion, the final results tend to be improved given that all sources are informative and complementary to one another. Section 4.5 provides some analysis on the failure case, which is unfortunately somewhat superficial.

**Summary Of The Paper:**

This work addresses the challenge of prompt engineering under few-shot setting. The key idea is to treat conventional zero-shot CLIP transfer inference as general knowledge, and enforce that the learned prompt will not diverge much from such general knowledge. In implementation, the gradient at each optimization step will be determined according to the divergence of two gradient direction. Comprehensive experiments are conducted to validate the effectiveness of the proposed idea.

**Summary Of The Review:**

Overall I would regard this is an interesting albert simple work toward more robust prompt engineering, under the usual setting of few-shot samples. It will be a good addition to the conference.

---

> ### Author Response · Authors · 2022-11-15
> **Response to reviewer Wdn9**
>
> **Q4.1** The proof of generalization bound is not much informative. It is a direct extension of the theoretical results in prior works (e.g. Zhang and Ye's 2012 paper).
>
> **A4.1** We do not claim the theoretical analysis as our contribution. We provide the analysis to theoretically justify why aligning the update gradient to the general knowledge of the CLIP could bring performance improvement, which has not been investigated in existing works.
>
> ---
>
> **Q4.2** More insights are required for the readers fully understanding the empirical superiority of the proposed idea. In specific, the "general knowledge" induced by standard zero-shot CLIP transfer does not provide ground truth. The gradient-calculating rule as proposed in Eq. 4 is essentially a fusion of two kinds of cues: one is from zero-shot extension, and the other is data-driven. The trick is to cleverly choose the way for the fusion. In the practice of multi-source fusion, the final results tend to be improved given that all sources are informative and complementary to one another. Section 4.5 provides some analysis on the failure case, which is unfortunately somewhat superficial.
>
> **A4.2** Our ProGrad is not a fusion of the two sources. Instead, we treat the information from zero-shot prediction as a regularization at the gradient level rather than supervision/loss level when fune-tuning on the downstream samples. The fusion of the two sources is more like knowledge distillation which adds a KL divergence loss to cross-entropy classification loss. Our empirical results in Table 4 of the main paper shows that knowledge distillation promotes CoOp in low-shot (e.g., 1,2 and 4) but makes the performance drop when number of shots is larger than 4.
> ProGrad is based on an assumption that  pre-trained VLMs have already captured all the knowledge needed in the downstream task and the goal is to compose a domain-specific query. Therefore, our regularization strategy is to ensure that each updated step does not conflict with the pre-trained information. The failure cases in Section 4.5 demonstrate that the error of our ProGrad mainly stems from the imprecise zero-shot information.

---

### Official Review · Reviewer_7YcA · 2022-10-31

**Confidence:** 4
**Correctness:** 3
**Technical Novelty And Significance:** 2
**Empirical Novelty And Significance:** 2
**Recommendation:** 3

**Clarity, Quality, Novelty And Reproducibility:**

Clarity of Introduction is very low, but for method section it is decent.

Novelty is very low. The method is very incremental.

**Strength And Weaknesses:**

###STRENGTHS
1. Wide range of experiments

### WEAKNESSES
1. Very incremental contribution
- The delta over CoOP is very small. Infact the method is a very simple extension of CoOP. I don't know why the authors made a whole new name.
2. Results not convincing
- The reason is that the improvement over CoOP seems very negligible in more important datasets like ImageNet and Table2. Can you aadd a table which shows results on individual datasets?
- Some tables have ProGrad and others have ProGrad++. If ProGrad++ is also your method, add it into the method section.
- Standard deviation is missing. So it is very difficult to draw any conclusion. 0.27 improvement on ViT-B in Table 2 can easily go away with multiple runs.
3. Not properly motivated
- The method does not have a sound technical motivation. It seems like a heuristic.
- Phenomenon shown in Fig.2 is not enough. It could be a cherry-picked exception.
- Starting of sec3.2 should motivate the challenge with previous methods. However, it says that it has been discussed in Introduction. Can you discuss it properly?
4. Poor Writing
- Although the method section is decently written, the introduction is hard to follow.

**Summary Of The Paper:**

### Problem
The paper tackles the problem of prompt tuning for large scale vision-language models. This is a very recent direction.

### Proposed method
The method is built on top of (i) zero-shot CLIP with hand-crafted prompts and (ii) CoOP which proposed to fine-tune context vectors as tunable prompts. The method proposes a modification on CoOP where the gradients are slightly modified.

### Experimental validation
Authors perform experiments on commonly used datasets.

**Summary Of The Review:**

I am voting for rejection because:
1. The delta over CoOp is very incremental.
2. The method seems like a heuristic without any proper technical motivation. Even more heuristics have been added into experiment section which are missing from method section. It is not enough for a conference paper.
3. The improvement over CoOP is small and standard deviation has not been reported. This makes it difficult to draw a conclusion.


=======================
POST REBUTTAL
=======================
After carefully going through the authors comments and the other reviews, I have decided to keep my rating the same. My reasons remain the same, which have not been addressed in the authors response:
1. The method is an incremental extension of CoOP. The fact that the authors have done serious re-branding is not so convincing. If it is a simple extension, saying so is something I would have preferred.
2. Standard deviation is missing from tables. It is good Machine Learning practise to have this. Despite me asking for this, the authors instead of suggesting that 'we will add this to the final version....', the authors chose to disagree with it.
3. I felt that the introduction is not well written. Authors 'disagree with it, saying that other reviewers did not feel this way, so my claim is wrong'. There are multiple reviewers to get different opinions.
4. I still feel that the motivation of the method is heuristics.

While these factors might be small alone, but all of them combined together make it a weak paper in the current format. These are changes that the authors can do to make the paper stronger. But the authors did not even accept these weaknesses, let alone proposing to work on them in the final version. Therefore, it is a reject from my side.

---

> ### Author Response · Authors · 2022-11-15
> **Response to reviewer 7YcA (1/2)**
>
> **Q3.1** Very incremental contribution. The delta over CoOp is very small. In fact the method is a very simple extension of CoOp.
>
> **A3.1** We disagree with both of the points. This judgement also conflicts with other reviewers. We would appreciate it if the reviewer could provide more detailed evidence so that we can help more to address the mixed reviews.
>
> **Contribution**: Our main contributions are as follows: (1) (research problem) We proposed an underexplored research question about how to adapting vision-language models with prompt tuning with *extremely few-shot* (e.g., 1 shot) training samples in the downstream domain. (2) (method) We proposed a training framework, ProGrad, which aligns the gradient of the prompt to the general knowledge to prevent prompt tuning from forgetting the knowledge learned from the CLIP model. (3) (experimental analysis) We conduct extensive experiments on 15 datasets for three tasks (few-shot classification, domain generalization, and base-to-new generalization).
>
> **Difference with CoOp** Our ProGrad is orthogonal to CoOp rather than its simple extension, and ProGrad can work as a plug-in to not only CoOp but also CoCoOp. First, as for motivations, CoOp focused on optimizing context in prompts instead of prompt engineering, while we focused on preventing forgetting the knowledge of the foundation models during the prompt tuning process. Second, as for technical implementations, CoOp is a plain cross-entropy trained model, while our ProGrad updates the gradients which are aligned to the general knowledge. As a result, CoOp negatively affects the generalization ability of the original pre-trained model, while our ProGrad not only improves the downstream performance (Section 4.2) but also achieves higher generalization power of the foundation model (Section 4.3, 4.4, D.1).
>
> Note that other reviewers (edmg, VFpS and Wdn9) acknowledged the novelty of our method, e.g., ``The proposed method looks novel and makes much sense'' by edmg, ``The idea is novel in the context of soft-prompt tuning'' by VFpS, ``The idea is novel, investigating of CLIP's few-shot generalization from an unexplored aspect'' by Wdn9.
>
> ---
>
> **Q3.2** Can you add a table which shows results on individual datasets?
>
> **A3.2** Actually, we have reported the results in the original paper. Please find them in Table 11, Table 12 and Table 13.
>
> ---
>
> **Q3.3** Some tables have ProGrad and others have ProGrad++. If ProGrad++ is also your method, add it into the method section.
>
> **A3.3** The difference between ProGrad and ProGrad++ is that ProGrad++ further adapts prompt ensemble. For fair comparisons, we mainly reported the performance of ProGrad based on single prompt. Following your suggestion, we further elaborated it in the revised paper Section 4.2.
>
> ---
>
> **Q3.4** Standard deviation is missing. So it is very difficult to draw any conclusion. 0.27 improvement on ViT-B in Table 2 can easily go away with multiple runs.
>
> **A3.4** We disagree. Actually, we have reported the standard deviation in Section D.2 in our original version.
> Our ProGrad achieves outperforms baselines by large margins on many settings: (1) significant improvement when given extremely few-shot samples, e.g., in Figure 4, ProGrad outperforms CoOp by 9.5\%, 6.9\% and 5.1\% on FGVCAircraft, EuroSAT and
> Flowers102 given 1 shot, and the average improvement over 11 datasets is 3.2\%.
> (2) shows significant improvement on domain generalization, e.g., in Table 2(a), ProGrad beats CoOp by 1.23\%, 0.99\%, 2.1\%, 0.98\% and 3.17\% on ImageNet, ImageNet-V2, ImageNet-Sketch, ImageNet-A and ImageNet-R. Such improvement is noteworthy as ImageNet and its variants are challenging dataset with 1000 classes.
> (3) outperforms baselines by a large margin in base-to-new generalization, e.g., in Table 3, ProGrad brings 4.7\% improvements on unseen classes.
>
> We did not find the mentioned 0.27 improvement. We would appreciate it if the reviewer could be more specific.
>
> ---
>
> **Q3.5** The method does not have a sound technical motivation. It seems like a heuristic.
>
> **A3.5** Our technical motivation is to align the gradient of the prompt to the pre-trained general knowledge to prevent prompt tuning from forgetting.
> We also justify the effectiveness of our method from generalization error analysis which is presented in Section 3.2. By virtue of Rademancher Complexity, we show that, under the assumption the gap between pre-trained and downstream domains is small, our ProGrad model achieves a much lower bound that CoOp.

---

> ### Author Response · Authors · 2022-11-15
> **Response to reviewer 7YcA (2/2)**
>
> **Q3.6** Phenomenon shown in Fig.2 is not enough. It could be a cherry-picked exception.
>
> **A3.6** Fig.2 is to intuitively demonstrate in few-shot scenarios, prompt tuning might learn spurious feature to predict. It is not a cherry-picked exception, and has been validated by many works [1-3].
>
> [1] Geirhos et al. Shortcut Learning in Deep Neural Networks. Nature Machine Intelligence
>
> [2] Yue et al. Interventional Few-Shot Learning, NeurIPS 2021.
>
> [3] Kang et al. Relational Embedding for Few-Shot Classification. ICCV 2021
>
> ---
>
> **Q3.7** Starting of Sec. 3.2 should motivate the challenge with previous methods. However, it says that it has been discussed in Introduction. Can you discuss it properly?
>
> **A3.7** The first paragraph in Section 3.2 of the original paper has adequately introduced the challenges and our motivation in details. Specifically, ``As we introduced in Section 1, CoOp faced a challenge that the transfer performance drops when the number of annotations is very limited (e.g., one per class), even underperforms the zero-shot transfer. Also, CoOp heavily relies on anti-overfitting techniques such as early stopping and data augmentation. To overcome the over-fitting challenge, we propose an effective and efficient fine-tuning paradigm ProGrad to align the few-shot downstream knowledge with the large-scale general knowledge.''
>
> ---
> **Q3.8** Poor writting
>
> **A3.8** This statement conflicts with all the other reviewers, who unanimously agreed this paper is clearly written and easy to follow. We would appreciate it if the review could be more specific on in which part the writing quality is not satisfying.

---

> ### Author Response · Authors · 2022-12-04
> **Looking forward to your feedback**
>
> Thank you again for your time and effort in our submission! Just a kind reminder that the discussion period is enabled through Dec 12. We would really appreciate it if you could share your further thoughts on our submission. Please do not hesitate to reply to us if you have any further questions and concerns, and we are happy to provide further clarification!

---

### Official Review · Reviewer_VFpS · 2022-10-31

**Confidence:** 4
**Correctness:** 3
**Technical Novelty And Significance:** 2
**Empirical Novelty And Significance:** 2
**Recommendation:** 6

**Clarity, Quality, Novelty And Reproducibility:**

The work's quality, clarity, and originality are satisfactory to me. As stated, the formulation to prevent forgetting is not new; however, in the context of soft-prompt tuning, no one has previously considered it. Importantly, the results demonstrate that this formulation actually aids according to various evaluation protocols.

**Strength And Weaknesses:**

**Strength:**
- Paper is clearly written and organized.
- Theoretical justification and derivation are correct as far as I know unless other reviewers raise important concerns. For me, this paper address the same question as [2,3] did in order to improve generalization. I would suggest authors to elaborate these similar papers in their related work.
- The idea is novel in the context of soft-prompt tuning, but it has been already explored in multi-task learning and continual learning [1]. It would be even nice if authors discuss a bit more about this prior work or similar ones in the related work.
- Ablations and experiments are complete in the main body of the paper as well as the appendix part.

**Weakness:**

According to my experience working with **CoOp** and **CoCoOp**, the two baselines of this paper, in CoCoOp work, all results are reported on **4 learnable tokens** initialized with "A photo of a {class}",  **16 shots** and **ViT/B-16 backbone** on three different random seeds for three different tasks: **base-to-new-generalization**, **cross-dataset transfer learning**, and **domain generalization**. Regarding this information, I would appreciate if the authors answers following questions:

1. Do authors train all baseline models themselves or they adopt the results from the main paper?
2. Are learnable prompts initialized with "A photo of a {class}" or they are trained from scratch?
3. I would ask authors why the number of shots are different in tasks  **base-to-new-generalization** and **domain generalization** in Table 1 and 2. Are there any specific reasons? Can they provide the performance with same hyper-parameters? I would suggest authors to be consistent with the baselines in terms of hyper-parameters to conclude a fair comparison. To me, Table 1 and Table 2 looks a bit wired as you change the training hyper-parameters. Maybe for 16 shots, CoCoOp performs better.
4. I noticed that authors did not provide results for cross-dataset transfer learning (Table 2 of CoCoOp paper). I would suggest to report these numbers as well since it makes the paper more complete.
5. Regarding ProGrad++, where authors use the idea of prompt ensemble, would it be possible to clarify how the prompt design changes in terms of number of learnable tokens. To me, since the input prompts changes, the length of learnable prompt would change accordingly and it is a bit unclear how many learnable parameters exist.



[1]. Farajtabar et al., Orthogonal Gradient Descent for Continual Learning, 2019

[2]. Lu et al. Prompt Distribution Learning, CVPR 2022

[3]. Derakhshani et al., Variational Prompt Tuning Improves Generalization of Vision-Language Models, Arxiv 2022

**Summary Of The Paper:**

This work highlights an intriguing problem regarding the tuning of soft prompts in vision and language models. This research asserts that current soft-prompt tuning papers have a tendency to overfit on training data while losing generalization ability on test data. To avoid this, they strive to optimize soft prompts in directions so that they do not clash with general information encoded in vision and language models. This eliminates contradictory directions and helps prevent forgetting. The method proposed in this paper is not novel in general and has been explored in multi-task learning and continual learning. However, this is the first time it is used for soft prompt tuning and I believe it is a appropriate contribution and is aligned with ICLR. In terms of experiments and comparisons, I think it is thorough and authors evaluate their model from different view points.

**Summary Of The Review:**

This paper addresses the issue of reducing the over-fitting problem for downstream adaptation of vision and language models as a whole. I vote for "marginally above the acceptance threshold," unless other reviewers indicate critical issues and I will decrease or increase my score in case it is needed.

---

> ### Author Response · Authors · 2022-11-15
> **Response to review VFpS (1/2)**
>
> **Q2.1** Discuss some prior papers in the related work.
>
> **A2.1** Thanks for your suggestion. We have updated the related work (Section 2) in the revised version.
> (1) Comparison with ProDA and VPT. Prompt tuning for VLMs suffers from generalization degradation problem in few-shot adaptation, our ProGrad, ProDA and VPT share the same goal to tackle such important issue.
> ProGrad bridges the generalization gap by aligning the gradient of the prompt to the general knowledge of CLIP model.
> Unlike ours, ProDA and VPT resort to distribution learning strategy to improve generalization robustness.
> ProDA adapts VLMs to downstream classification tasks by learning a prompt distribution over the output embedding space. VPT introduces variational prompt tuning by combining a base learned prompt with a residual vector sampled from a instance-specific underlying distribution.
> (2) Comparison with ``Orthogonal Gradient Descent for Continual Learning''.  OGD projects the gradients from new classes to the orthogonal direction of the gradients of previous tasks. However, as we have no access to pre-training
> process, the requirement of OGD to store the gradients of old tasks is not possible for prompt tuning. Moreover, OGD alters the gradients of downstream tasks even in non-conflicting scenarios which might result in sub-optimal performance for downstream tasks.
> ---
> **Q2.2** Do authors train all baseline models themselves or they adopt the results from the main paper?
>
> **A2.2** We re-trained all the baseline models by ourselves using their released code from CoOp and CoCoOp for fair comparisons.
>
> ---
> **Q2.3** Are learnable prompts initialized with "A photo of a \{class\}" or are they trained from scratch?
>
> **A2.3** All learnable prompts are initialized from hand-crafted prompts for all prompt-based model, i.e., CoOp, CoCoOp and ProGrad. To be specific, for few-shot classification (Section 4.2), we use the hand-crafted prompts listed in Table 6 of Appendix (we prepend the hand-crafted prompts with zeros if their length is less than the token length 16); for domain generalization and base-to-new generalization (Section 4.3 and Section 4.4), since the learnable token length is 4, we initialize all the prompts as ``a photo of a ''. More implementation details for initialization can be found in Appendix B.
>
> ---
> **Q2.4** I would ask authors why the number of shots are different in tasks base-to-new-generalization and domain generalization in Table 1 and 2. Are there any specific reasons? Can they provide the performance with same hyper-parameters? I would suggest authors to be consistent with the baselines in terms of hyper-parameters to conclude a fair comparison.
>
> **A2.4** We understood the reviewer has a concern that the comparison results of CoOp and CoCoOp differ from the reported results in the CoCoOp paper. We argue that CoCoOp made **an unfair comparison** with CoOp on base-to-new-generalization and domain generalization tasks. The CoCoOp paper sets the number of training epochs to 200 (except for ImageNet where the number is fixed as 50) for CoOp, while all experiments of CoCoOp are run only for 10 epochs. Differently, we set same training epochs (i.e., 100 epochs for 4 shots) for all methods to make a fair comparison.
>
> Using different training strategies for different methods makes an unfair comparison: **extremely less training epochs work as early stopping and yield much better performance on unseen classes**, which is why CoCoOp performs worse in our Table 3 as we make its training epochs identical to CoOp. Therefore, we intent to set same training epochs for all methods to make a fair comparison.
>
> *Reason to use 4 shots training:* We consider extremely few-shot scenarios to better evaluate the robustness and generalization of methods. Therefore, we focus more on the 4-shot setting rather than the original 16-shot setting.
>
> Although both CoOp and CoCoOp can surpass the zero-shot CLIP in the domain generalization setting for 16 shots, our ProGrad is the only prompt tuning method that yields higher performance than zero-shot CLIP for 4 shots. These observations indicate that our ProGrad has a stronger generalization ability when the training samples in the downstream domain are extremely limited.
>
> ---
> **Q2.5** To me, Table 1 and Table 2 looks a bit wired as you change the training hyper-parameters. Maybe for 16 shots, CoCoOp performs better
>
> **A2.5** To further address your concerns, we conduct the experiments in Table 1 and Table 2 in the revised paper under 16 training shots and set training epoch to 10 for CoCoOp (same as the original CoCoOp setting). The results are shown in the following tables, where the conclusion remains the same: our ProGrad exhibits stronger domain generalization and unseen classes generalization power.

---

> > ### Comment · Reviewer_VFpS · 2022-12-03
> > **Response to the authors**
> >
> > Thanks for the response. It was informative and complete. I will keep my score unchanged.

---

> > > ### Author Response · Authors · 2022-12-04
> > > **Thank you for the acknowledgement of our responses**
> > >
> > > Thank you for the feedback that our response was informative and complete and the score will be kept positive. Please feel free to let us know if you have any further questions or concerns during the discussion period.

---

> ### Author Response · Authors · 2022-11-15
> **Response to review VFpS (2/2)**
>
> **A2.5**
> |                  | ImageNet       | -V2            | -Sketch        | -A             | -R             |
> |------------------|----------------|----------------|----------------|----------------|----------------|
> | CLIP             | 66.73          | 60.83          | 46.15          | 47.77          | 73.96          |
> | CoOp             | 71.51          | 64.20          | 47.99          | 49.71          | 75.21          |
> | CoCoOp           | 71.02          | 64.07          | 48.75          | 50.63          | 76.18          |
> | ProGrad | **71.74** | **64.53** | **48.94** | **50.92** | **76.38** |
>
> *Title: Domain generalization results based on ViT-B/16 with 16 shots*
>
> |         | Base  | New   | Harmonic |
> |---------|-------|-------|----------|
> | CLIP    | 69.34 | 74.22 | 71.70    |
> | CoOp    | 82.69 | 63.22 | 71.66    |
> | CoCoOp  | 80.47 | 71.69 | 75.83    |
> | ProGrad | 81.71 | 74.42 | 77.89    |
>
> *Title: Averaged accuracy (\%) over 11 datasets for base-to-new generalization with ViT-B/16 backbone and 16 shots.*
>
> ---
> **A2.6** I noticed that authors did not provide results for cross-dataset transfer learning (Table 2 of CoCoOp paper). I would suggest to report these numbers as well since it makes the paper more complete.
>
> **Q2.6** Thanks for the suggestion. We have added the results of cross-dataset transfer learning in the revised version (Section D.1). Specifically, our ProGrad not only achieves the highest performance on the source dataset ImageNet, but also outperforms other baselines 9 out of 10 target datasets. In specific, ProGrad shows a gain of +3.57\%, +3.27\%, +4.22\% on Caltech101, Standford Cars and SUN397 datasets over CoCoOp. The results are listed in the following table.
>
> |           | ImageNet     | Caltech101   | OxfordPets   | StanfordCars | Flowers102   | Food101      | FGVCAircraft | SUN397       | DTD          | EuroSAT      | UCF101       | Average      |
> |-----------|----------------|----------------|----------------|----------------|----------------|----------------|----------------|----------------|----------------|----------------|----------------|----------------|
> | CoOp      | 61.34          | 84.48          | 85.99          | 54.16          | 60.10          | 75.48          | 14.09          | 57.48          | 35.32          | **26.72** | 57.56          | 55.70          |
> | CoCoOp    | 61.04          | 84.73          | 86.42          | 52.34          | 61.24          | 73.79          | 13.74          | 55.94          | 36.60          | 23.46          | 57.97          | 55.21          |
> | ProGrad | **62.17** | **88.30** | **86.43** | **55.61** | **62.69** | **76.76** | **15.76** | **60.16** | **39.4** | 24.87          | **58.70** | **57.36** |
>
> ---
> **A2.7** Regarding ProGrad++, where authors use the idea of prompt ensemble, would it be possible to clarify how the prompt design changes in terms of number of learnable tokens? To me, since the input prompts changes, the length of learnable prompt would change accordingly and it is a bit unclear how many learnable parameters exist.
>
> **Q2.7** We only use the hand-crafted prompt ensembling to generate $G_g$ for ProGrad++, which provides a more accurate general direction. Therefore, we still use a single prompt with 16 learnable tokens for prompt tuning. We also updated the description of ProGrad++ in the revised version (Section 4.2).

---

### Official Review · Reviewer_edmg · 2022-11-01

**Confidence:** 3
**Correctness:** 4
**Technical Novelty And Significance:** 3
**Empirical Novelty And Significance:** 2
**Recommendation:** 6

**Clarity, Quality, Novelty And Reproducibility:**

- Clarity: The paper is easy to follow.
- Quality: Its claim is well supported
- Novelty: The proposed ProGrad is novel
- Reproducibility: The method is clearly described, and its hyperparameters are provided.

**Strength And Weaknesses:**

- Strength
1. The proposed method looks novel and makes much sense. Using a generalizable prompt to regularize the learning of a domain-specific prompt is a simple and effective strategy to avoid overfitting in few-shot learning.
2. The experiment is comprehensive. It includes three aspects: standard few-shot image classification, domain generalization, and unseen class classification. The comparison to KD (Table 4) and the cosine classifier (Table 5) directly support its claims that its regularized gradient direction is the key to improving the results.
- Weaknesses
1. The improvement of ProGrad looks marginally better and may be affected by the choice of hyperparameters. What is the upper bound result this approach can get? How about simulating the upper-bound case using many training data to get an optimized prompt instead of using the man-made prompt to create G_g?

**Summary Of The Paper:**

This work addresses VLM few-shot learning with prompt tuning. It addressed the overfitting of the learned prompts by regularizing the direction of gradients. The direction is enforced not in the opposite direction of the gradient computed with the human-selected prompt. The paper provides a theoretical justification to show its regularized gradients lead to a smaller error bound. The experiments in few-shot image classification, domain generalization, and unseen class classification show that the proposed method achieves better accuracy in all benchmarks.

**Summary Of The Review:**

The method is well-motivated and described. Its experiments cover a wide range of settings and show improvements. Overall it looks like a well-executed paper.

---

> ### Author Response · Authors · 2022-11-15
> **Response to reviewer edmg**
>
> **Q1.1** The improvement of ProGrad looks marginally better and may be affected by the choice of hyperparameters.
>
> **A1.1** Overall, our ProGrad achieves (1) significant improvement when given extremely few-shot samples. For example, in Figure 4, ProGrad outperforms CoOp by 9.5%, 6.9% and 5.1% on FGVCAir- craft, EuroSAT and Flowers102 given 1 shot, and the average improvement over 11 datasets is 3.2%. (2) outperforms baselines by a large margin in base-to-new generalization. For example, in Table 3, ProGrad brings 4.7% improvements on unseen classes. (3) shows significant improvement on domain generalization. For example, in Table 2(a), ProGrad beats CoOp by 1.23%, 0.99%, 2.1%, 0.98% and 3.17% on ImageNet, ImageNet-V2, ImageNet-Sketch, ImageNet-A and ImageNet-R. Such improvement is noteworthy as ImageNet and its variants are challenging dataset with 1,000 classes.
>
> ---
>
> **Q1.2** What is the upper bound result this approach can get? How about simulating the upper-bound case using many training data to get an optimized prompt instead of using the man-made prompt to create $G_g$?
>
> **A1.2** Following edmg’s suggestions to estimate the upper bound, we first optimize a prompt with the vanilla cross-entropy loss on the full downstream dataset to obtain $G_g^\text{full}$, then use $G_g^\text{full}$ instead of one offered by hand-craft prompt to train the ProGrad. The results are shown in the following tables. The results indicate that a more optimized general knowledge direction $G_g^\text{full}$ can elicit a stronger ProGrad model. We also added the results in the revised appendix (Section D.2).
>
> | \#shots               | 1     | 2     | 4     | 8     | 16    |
> |-----------------------|-------|-------|-------|-------|-------|
> | ProGrad               | 88.68 | 87.98 | 89.99 | 89.99 | 92.17 |
> | ProGrad$_\text{full}$ | 91.08 | 91.70 | 91.76 | 91.84 | 92.86 |
>
> *Title: Upper-bound results of few-shot learning on Caltech101.*
>
> | \#shots               | 1     | 2     | 4     | 8     | 16    |
> |-----------------------|-------|-------|-------|-------|-------|
> | ProGrad               | 46.14 | 49.78 | 54.43 | 60.69 | 63.97 |
> | ProGrad$_\text{full}$ | 66.11 | 67.04 | 67.24 | 68.46 | 69.27 |
>
> *Title: Upper-bound results of few-shot learning on DTD.*
>
> | \#shots               | 1     | 2     | 4     | 8     | 16    |
> |-----------------------|-------|-------|-------|-------|-------|
> | ProGrad               | 58.38 | 61.81 | 65.62 | 69.29 | 73.75 |
> | ProGrad$_\text{full}$ | 70.54 | 71.57 | 71.66 | 72.73 | 75.27 |
>
> *Title: Upper-bound results of few-shot learning on Stanford Cars.*

---

### Author Response · Authors · 2022-11-15
**Response to All Reviewers**

Dear Program Chair, Senior Area Chair, Area Chair, and Reviewers,

First of all, we gratefully thank all the reviewers for their thoughtful comments and feedback. In this paper, we found the conventional prompt tuning may damage the generalization ability of foundational vision-language-pretrained (VLM) model, e.g., CoOp trained with extremely few- shot samples may even under-perform the zero-shot CLIP prediction. We are encouraged that they find our proposed method, ProGrad which aligns the gradient of the prompt to the general knowledge to prevent prompt tuning from forgetting the knowledge learned from the CLIP model, is novel (edmg, VFpS, Wdn9), simple (edmg, Wdn9, 6JR9), effective (edmg, Wdn9, 6JR9) and makes much sense (edmg). We are glad that reviewers find our paper is easy to follow (edmg), clearly written and organized (VFpS, Wdn9, 6JR9) an interesting work (Wdn9) and has done comprehensive experiments (edmg, VFpS, 7YcA).

In addition, we have uploaded a new revision of our paper (modifications are in blue text), in which we incorporate most of the comments from the reviewers. Specifically, we:
1. Added the comparison with related works ProDA, VPT and OGD in Section 2.
2. Elaborated the implementation of ProGrad++ (ensemble version of our ProGrad) in Section 4.2.
3. Added the cross-dataset transfer experiments in Section D.1 in the appendix.
4. Reported the upper-bound performance in Section D.2 in the appendix.

As our paper received mixed ratings, i.e., four 6 (positive) and one 3 (negative), it would be appreciated if the reviewers could have a look at our responses and revision. We have tried our best to address your concerns in our responses in detail. Hope that our responses answered the questions. Please let us know at your early convenience if you have further questions or concerns.

Best regards,

Authors of Paper #4983

---

### Author Response · Authors · 2022-12-20
**Irresponsible Review from Reviewer 7YcA**

Dear Program Chairs, Senior Area Chairs and Area Chairs,

We’d like to thank you for reviewing our paper. However, there is one misconducted reviewer 7cYA for which we seek your attention.

**First, reviewer 7YcA shows a clear disrespect to our work.**

We are disappointed to receive his comment “the authors have done serious re-branding''. Identifying our work as ``re-branding’’ is very unprofessional, which is almost an accusation of “plagiarism”. We want to point out that such disrespect of our academic integrity without valid evidence is a violation of ICLR code of conduct. As acknowledged by all the other reviewers, our work is orthogonal to CoOp (which he refers as the “brand”), which works as a plug-in to not only CoOp but also other prompt tuning methods like CoCoOp.

**Second, reviewer 7cYA deliberately avoids discussion**

He was not engaged in author-reviewer discussion session: he updated his review **two days after** the author-reviewer discussion period, which did not give us a chance to reply. In particular, he edited the comment rather than add a new one, so we did not get any notification.

**Third, reviewer 7cYA deliberately ignores the fact**

About his comment  ``Standard deviation is missing from tables.’’. Actually, we provided the standard deviation information in Appendix in the original paper (Table 9, 10) and the updated paper (Table 11, 12), and we mentioned them in the main paper and the rebuttal.  Unfortunately, he ignored our response in the post-rebuttal review.

**Fourth, reviewer 7cYA deliberately provides uninformative review.**

He regarded our paper as ``poor writing’’ without any specification. As all the other reviewers unanimously acknowledged our good writing, he argued that it is just because different reviewers have different flavors. We are shocked by such an unprofessional misconduct.

In summary, we think that reviewer 7cYA deliberately rejects our work: (1) showed no respect to our work and regarded it as ``re-branding’’ (2)  only updated the review after the discussion session and did not give us a chance to reply (3) chose to ignore the results that we have already provided in the original and updated paper (4) judged the writing of our paper but is unwilling to specify. We’d appreciate it if the AC can disregard the review of 7cYA.

---

### Decision · Program_Chairs · 2023-01-20

**Decision:**

Reject

**Justification For Why Not Higher Score:**

Multiple reviewers raise concerns about the experimental results. For example, the result improvements aren't very clear compared to CLIP, CoOp, and CoCoOp, as not all the results are included and compared in the paper (in the author's response, e.g., A 2.5, the improvement over CoCoOp seems pretty marginal).

**Justification For Why Not Lower Score:**

N/A

**Metareview: Summary, Strengths And Weaknesses:**

This paper proposes a new prompt learning method, Prompt-aligned Gradient, for more generalizable prompt tuning under few-shot learning scenarios. In particular, it addresses the overfitting issue of the learned prompts by regularizing the direction of gradients. The method is evaluated on multiple image classification datasets across three different tasks: base-to-new-generalization, cross-dataset transfer learning, and domain generalization. Although the technical idea has been explored in continual learning in the vision and language research area, it is new in the context of prompt learning. However, the result improvements aren't very clear compared to CLIP, CoOp, and CoCoOp, as not all the results are included and compared in the paper (in the author's response, e.g., A 2.5, the improvement over CoCoOp seems pretty marginal). Reviewers have concerns regarding incremental contributions over CoOp, unconvincing results, etc.